# Thin, soft, wearable system for continuous wireless monitoring of artery blood pressure

Jian Li [1,2,8], Huiling Jia [1,2,8], Jingkun Zhou [1,2,8], Xingcan Huang [1,8], Long Xu[3], Shengxin Jia [1,2], Zhan Gao[1], Kuanming Yao [1], Dengfeng Li [1,2], Binbin Zhang[1,2], Yiming Liu [1], Ya Huang [1,2], Yue Hu[1], Guangyao Zhao[1], Zitong Xu[1], Jiyu Li[1,2], Chun Ki Yiu[1,2], Yuyu Gao[1], Mengge Wu [1,4], Yanli Jiao [1,2], Qiang Zhang[1], Xuecheng Tai [2,5], Raymond H. Chan[1,2], Yuanting Zhang [1,2], Xiaohui Ma[6] ✉ & Xinge Yu [1,2,7] ✉

Continuous monitoring of arterial blood pressure (BP) outside of a clinical setting is crucial for preventing and diagnosing hypertension related diseases. However, current continuous BP monitoring instruments suffer from either bulky systems or poor user-device interfacial performance, hampering their applications in continuous BP monitoring. Here, we report a thin, soft, miniaturized system (TSMS) that combines a conformal piezoelectric sensor array, an active pressure adaptation unit, a signal processing module, and an advanced machine learning method, to allow real wearable, continuous wireless monitoring of ambulatory artery BP. By optimizing the materials selection, control/sampling strategy, and system integration, the TSMS exhibits improved interfacial performance while maintaining Grade A level measurement accuracy. Initial trials on 87 volunteers and clinical tracking of two hypertension individuals prove the capability of the TSMS as a reliable BP measurement product, and its feasibility and practical usability in precise BP control and personalized diagnosis schemes development.

Cardiovascular diseases (CVDs), one of the leading causes of death worldwide, take an estimated of 18 million people's lives annually[1,2]. Blood pressure (BP) as an important biomarker provides remarkable insights into many hemodynamic parameters, such as stroke volume, heart rate, and cardiac output, which are closely relate to CVDs[3–5]. Monitoring of BP routinely can effectively reduce the attacking risk of CVD related diseases, such as sleep apnea, stroke, and coronary heart disease, since hypertension is a vital risk index during the development of CVD related diseases[6–9]. At present, clinics mainly rely on the cuff-based sphygmomanometers to measure the static BP, in which only

isolated casual systolic BP (SBP), diastolic BP (DBP) are recorded, and mean BP (MAP) can be calculated according to SBP and DBP. It is challenging to achieve continuous dynamic measurement of BP with current cuff-based sphygmomanometers due to the working mechanism of the cuff inflation[10–13]. In comparison, continuous BP monitoring not only provides an efficient monitoring method for patients with long-term or pre-diagnostic hypertension, but also makes a better prediction for CVDs related attacks in hypertensive individuals[5,11,14–17].

In clinic, the gold standard for continuous BP measurement is implanting an invasive fiber-based pressure sensor into the center of

[1]Department of Biomedical Engineering, City University of Hong Kong, Hong Kong, China. [2]Hong Kong Centre for Cerebro-Cardiovascular Health Engineering (COCHE), Hong Kong, China. [3]School of Mechanical and Aerospace Engineering, Jilin University, 130012 Changchun, China. [4]State Key Laboratory of Electronic Thin Films and Integrated Devices, School of Optoelectronic Science and Engineering, University of Electronic Science and Technology of China (UESTC), 610054 Chengdu, China. [5]Department of Mathematics, Hong Kong Baptist University, Hong Kong, China. [6]Department of vascular and endovascular surgery, The first medical center of Chinese PLA General Hospital, 100853 Beijing, China. [7]City University of Hong Kong Shenzhen Research Institute, 518057 Shenzhen, China. [8]These authors contributed equally: Jian Li, Huiling Jia, Jingkun Zhou, Xingcan Huang. ✉e-mail: maxiaohui@301hospital.com.cn; xingeyu@cityu.edu.hk

artery[18–20], which raises patients suffering and infection risk, and also too invasive for routine inspection and daily monitoring. In recent years, the rapid development of wearable electronics has made it possible to continuous monitor of some physiological indexes with wearable devices[21–23]. Therefore, continuous non-invasive BP monitoring with wearable devices has drawn great research attention. Several non-invasive methods, including optical based method (photo-plethysmography, PPG)[24–26], acoustic (ultrasound wall-tracking)[27,28], bioimpedance[8,9] and pressure sensor based tonometry[29–31], have been developed for this purpose. Among them, PPG is one of the most commonly used methods, however, which also suffers from signal aliasing from venous arteries and insufficient penetration depth (normally <8 mm). In addition, the complicated data processing system and the stringent setups for optical information processing further limit the development of PPG systems into real wearable device for continuous and long-term BP monitoring[32–34]. Ultrasound wall-tracking and bioimpedance have been considered as the promising candidates for continuous BP monitoring due to their high penetration depth and robust sensing capability in hemodynamic parameters[8,9,27,35]. However, to realize wearable/portable continuous monitoring of BP not only needs the devices that are designed in wearable manner, but also puts high demand on signal sampling and post processing for the BP calibration. Although wearable ultrasound transducers[27] and electrodes[8] have been developed for continuous BP measurement, high precise sampling equipment and bulky interface block their further application in continuous ambulatory BP monitoring. Tonometry that facilitates a pressure sensor to detect the artery deformation generated by blood propagation has been also considered for BP measurement due to its simple structure and low requirement on signal processing. While high sensitivity pressure sensors, including piezoresistive based[36,37] and capacitive based ones[38,39], all face the interfacial instability issue, and thus repeated calibration is required for BP evaluation.

Here, we report a class of materials, devices, mechanic designs, data processing methods and integration strategy for a thin, soft, miniaturized system (TSMS), that is capable for continuously monitoring of BP in real wearable format, with whose accuracy is comparable to the professional medical equipment. The TSMS composes three subsystems, including a sensing module to detect the blood pulse wave, an active pressure adaptation module to provide powerful back pressure for improved interfacial properties, and a data processing module to in-situ extract pulse transit time interval and transmit the measured data to a graphical user interface (GUI). The entire system only weighs 50 g, as thin as 4 mm, that is encapsulated in silicone as a wristband to provide friendly wearable interface for users. Highly chemical/physical stability of the piezoelectric material PZT 5H enables the sensor array with excellent uniformity and robustness while maintaining superior sensitivity due to its high piezoelectric coupling coefficient[40]. The active pressure adaptation unit is based on a build-in micro airbag together with a micro pump, which provides a powerful backpressure and close-looped feedback for the piezoelectric sensor array to effectively detect the arterial deformation generated by blood propagation. The measured blood pulse waveforms, and the time difference between their arrival at the two sensing units are preprocessed along with multiple extracted features and local pulse wave velocity (PWV) as input to a blood pressure estimation model. Compared to other cuffless BP monitoring devices (Supplementary Table 1), this work addresses the issues in system integration level, interfacial performance, and BP estimation model, thus showing great advantages in terms of wearability, continuance, dynamics, and most importantly, measurement accuracy (Grade A level), which will promote its application in continuous BP monitoring.

## Results
### Design of the device
Figure 1a shows the schematic illustration of the TSMS in a wearable wristband format, where the overall dimension is 15 cm in length, 35 mm in width and 4 mm in thickness. The TSMS includes three subsystems of a piezoelectric based pulse sensing system, an active pressure adaptation system and a data sampling/transmitting system (Fig. 1b, Supplementary Fig. 2). The pulse sensing system associates with a pair of sensors spacing 15 mm from each other, that adopts piezoelectric thin layers, lead zirconate titanate (PZT 5H) as the sensors (3 mm × 3 mm, 200 μm in thickness), as shown in Fig. 1b. The piezoelectric sensors are based on sandwiched top/bottom electrodes, where the top electrode is routed to the bottom substrate via a vertical interconnect access (VIA) for optimized mechanical robustness (Supplementary Fig. 3) and ease of electrical connection, which endows the sensors adaptation to various mechanical deformations (Fig. 1e, Supplementary Fig. 3). A bilayer of polyimide (PI, 25 μm thick) and gold (Au, 200 nm thick) that is designed in serpentine patterns to serve as the flexible electrodes and interconnects for the sensor array while realizing flexibility. (See methods for the details, Fig. 1b). The pair of the piezoelectric sensors is encapsulated with soft silicone (Poly-dimethylsiloxane, PDMS, 145 kPa), providing soft and conformal contact between the sensors and the artery located skin region, thus allowing for accurate conversion of local deformation of the sensor caused by the expansion/contraction of the artery into electrical outputs[41].

### Working principles and system integration
Piezo response is generated due to the potential difference across the sandwiched electrodes when suffering from mechanical deformation caused by blood propagation (Fig. 1a), and the piezo response is then transmitted to the data model, where it is converted into the continuous pulse wave based on a mathematic model (Supplementary Note 2) for the following features extraction. Meanwhile, by measuring the pulse travel time (PTT) of a single pulse waveform arriving at two sensors, and the distance between them, the local PWV can be calculated. The localized PWV together with the extracted multi physiological features (Supplementary Table 2) are then wirelessly transmitted to a pretrained data model implemented in remote server, wherein the continuous BP pattern is calculated based on the inputs. The predicted BP pattern is then transmitted to a mobile GUI, where continuous BP patterns of users are presented and recorded for further analysis.

The active pressure adaptation system associates with back-pressure generator, which composes of a micro pump (19 mm × 21 mm × 3.6 mm), a pair of soft silicone (Ecoflex, 120 kPa) based micro airbag array (Fig. 1b, f, Supplementary Fig. 4), and a silicone (Ecoflex, 120 kPa) one-way valve (Supplementary Fig. 5). The micro pump provides sufficient pressure for the pair of airbags (up to 40 kPa within 0.5 s), enabling its powerful support to the sensor array. The one-way valve at the outlet of the micro pump (Fig. 1b, Supplementary Fig. 6) acts as a pressure holder to maintain the pressure in the airbag as well as a damping valve to stabilize the active pressure adaptation system. Here, a multi-phase pumping method is employed (with a maximum pressure at 12 kPa under 5 pumping phases) to provide stable and appropriate support for the sensor array to detect the minor deformation in artery caused by blood propagation. The data sampling/transmitting system measures the piezo response and then processes the raw data into digital signal with a high precise 24-bit analogue-to-digital converter (ADC) before being transmitted to the micro-controller (MCU) (Fig. 1b). The data is then resampled by the MCU and transmitted to the mobile phone via the in-built Bluetooth Low-Energy (BLE) wirelessly. The wireless system could continuously work for almost 2 days (6 times measures per hour, and each measurement lasts 15 s to get continuous BP, Supplementary Fig. 7) powered by a coin-size rechargeable lithium-ion battery (80 mAh). All the electronic components, including the sensor array, the micro airbag array, the micro pump, the flexible printed circuit board (fPCB), and their interconnections (Fig. 1d) are encapsulated to form a flexible, lightweight (50 g) wristband, providing a totally wearable and comfortable

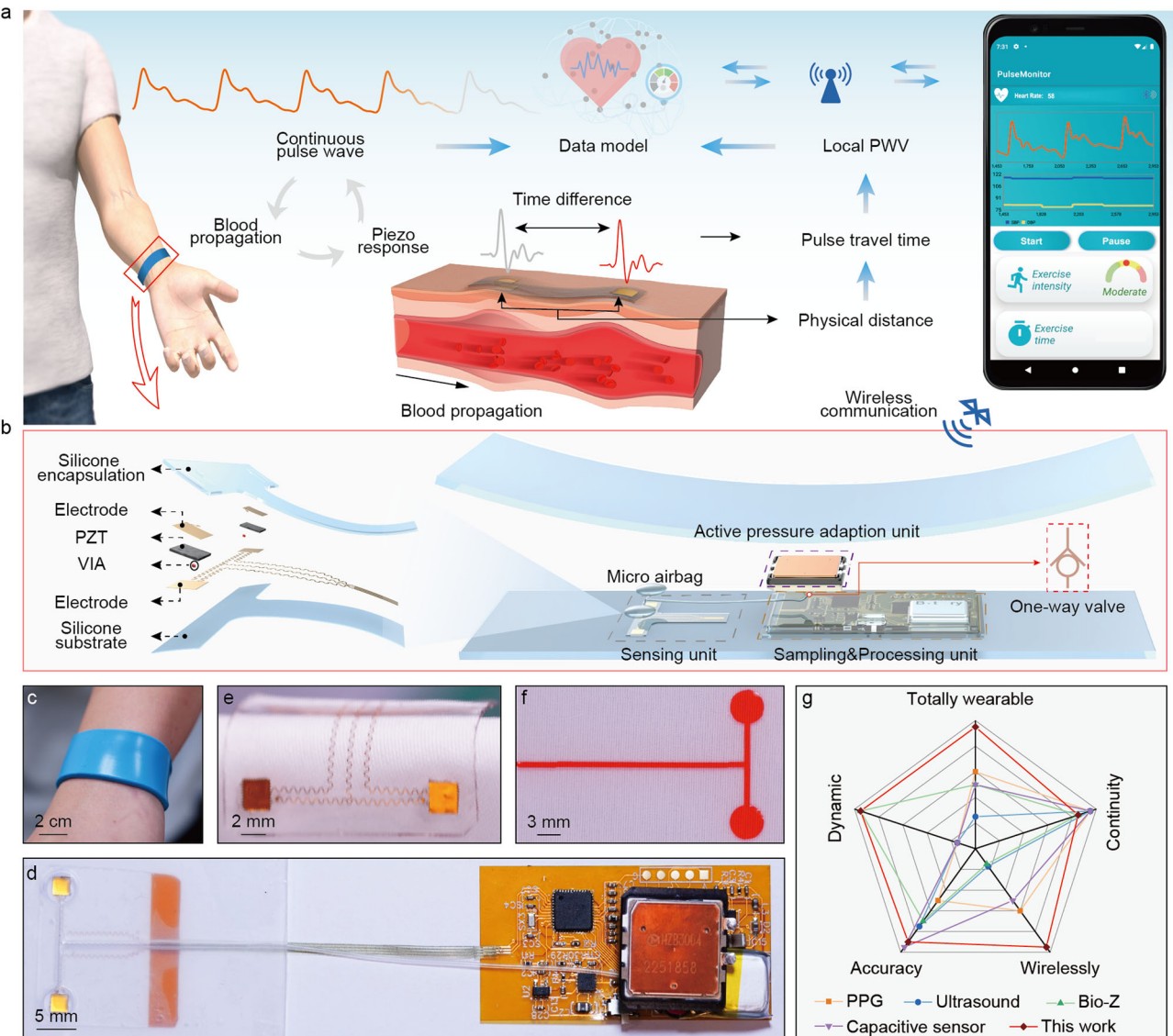

**Fig. 1 | Working principle and layouts of the wearable wireless continuous blood pressure monitoring system. a** Schematic diagram of signal conversion from piezo response to continuous blood pressure that is presented in a mobile graphic user interface (GUI). Physical distance between two sampling sites and time difference in two sensing units were utilized to calculate localized pulse wave velocity (PWV). Pulse wave features, together with localized PWV were transmitted to data for the estimation of beat-to-beat blood pressure (BP). **b** Explosive view of the wireless wristband, with three subsystems, sensing module, where the top surface and bottom surface of piezoelectric material lead zirconate titanate (PZT) are connected to top and bottom electrode, and top electrode is routed to bottom electrode by a vertical interconnect access (VIA), force generation module and signal processing module. **c** Optical image of the wireless wristband worn on user's wrist joint. **d** Optical image of all the system components before sealed in the silicone wristband. **e** and **f** Optical images of the sensor array and micro airbag array suffering from mechanical deformations. **g** Technical comparison between our device and published works utilizing Photoplethysmography (PPG), ultrasound wall tracking, bioimpedance (Bio-Z), and capacitive sensor for continuous BP monitoring in terms of wearability, accuracy, dynamics, continuance and wireless.

interface for users. The low power consumption of the TSMS keeps it from accumulating heat in the regional area. The temperature of the device maintains in a relatively low range (below 37 °C) after 1 h continuously working (Supplementary Fig. 8), and thus will not cause any thermal discomfort or safety concerns. To our best knowledge, the TSMS in this work providing a totally wearable, user-friendly interface is the most advanced continuous BP monitoring device in terms of wearability, accuracy, continuity, dynamics, and many other detailed aspects (Fig. 1g, Supplementary Table 1).

### Device characterization and piezo response conversion

The TSMS encapsulated by silicone in a wristband format provides an easy and efficient way for users to wear, where the sensor array can detect the mechanical pulse caused by blood propagation. While the use of PZT 5H as sensors is based on its excellent electrical, physical and chemical stabilities (Supplementary Figs. 9, 10), which also leads to interfacial contact issues with the skin due to its mechanical stiffness. Therefore, we designed an active pressure adaptation system, consisting of a pair of silicone based micro airbag (20 mm × 30 mm × 1.5 mm), a micro pump (19 mm × 21 mm × 3.6 mm) and a one-way valve, that can provide extra backpressure to the sensor array and thus maintain good contact behavior between the sensor and skin/artery. The adjustable backpressure of the airbags allows the sensors always to be able to accurately measure the pulse related piezo response (Fig. 2a). The pressure up to 40 kPa provided by the active pressure adaptation system guarantees the continuous and accurate monitoring of piezo response generated by blood pulse (Supplementary Fig. 11). The one-way valve, consisting of a perforated silicone hose, an

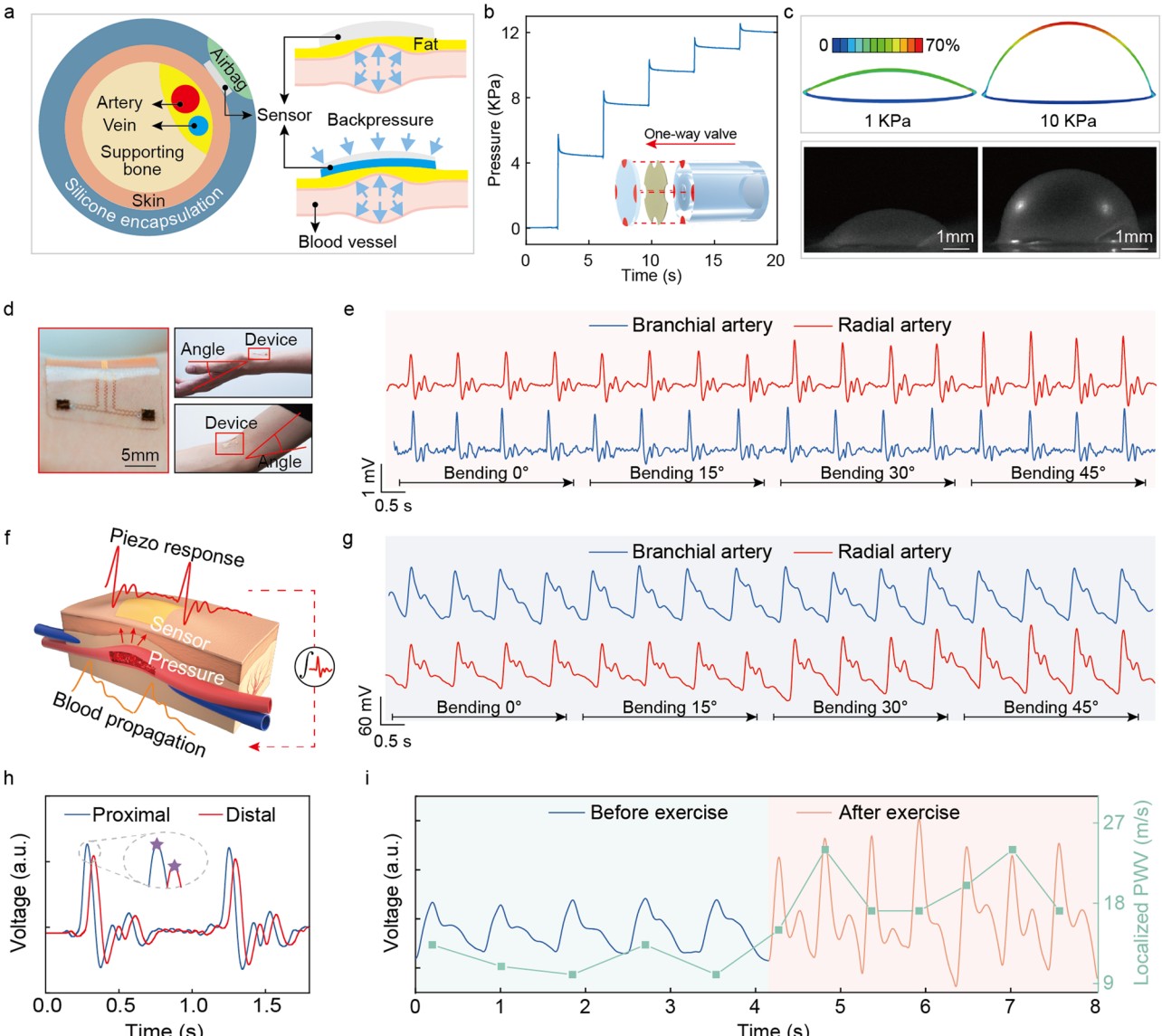

**Fig. 2 | Device characterization and signal analysis. a** Schematic illustration of the wireless wristband worn on the wrist, where the airbag provide backpressure to effectively increase the mechanical deformation of the sensor array generated from blood propogation. **b** The pressure in airbag under five pumping phases with pumping duration at 35 ms. With the one-way valve utilized, the pressure in the airbag maintains after the pump is turned off, and the pressure increase to 12 KPa after 5 pumping phases. **c** FEA results and corresponding optimal images present the coutour of the airbag under different inner pressure, 1 KPa and 10 KPa. **d** Optical images of the sensor array mounting on wrist and elbow joints under different deformation. **e** Measured peizo response of a sensing unit from radial artery and brachial artery under 0–45° bending deformations. **f** Schematic illustration of the blood propogation and generated piezo response. **g** Converted pulse waveform from the piezo response in (**e**) based the mathematical model illustrated in (**f**). **h** Illustration of the calculation of time difference between two sensing units from their piezo response. **i** Measured pulse wave on radial artery before and after exercise, and corresponding localized PWV.

isolation film, and a silicone cover, can effectively maintain the pressure in the airbags when the pump is turned off (Supplementary Fig. 5a). When the pump is turned on, there will be gaps between the bonding sites on the silicone cover and the silicone hose due to the pressure difference across the silicone cover (Supplementary Fig. 5b). In contrast, when the pump is off, the silicone cover will be pressed tightly against the silicone hose, maintaining a high pressure in the airbag side. With the one-way valve, the pressure in the airbags can maintain at a high level for a long time, with pressure decreasing to 30% of the original pressure after 1 h (Supplementary Fig. 5c–g). It is worthing noting that, although the pressure response time increases with the one-way valve equipped owing to increased system damping, the pressure (up to 40 KPa) can still be applied rapidly in <500 ms (Supplementary Fig. 6). While the increased system damping greatly enhances system stability, with less pressure fluctuations.

A control strategy of the active pressure adaptation system associated with multi-pumping phases was proposed for the purpose of intelligently operating the system. The pressure adaptation module serves as a powerful support to the sensor for increasing the sensing signal quality. However, too high back pressure level may also cause discomfort to users due to the closure of artery. Therefore, we set the maximum pressure at 12 kPa (90 mmHg) to avoid discomfortable feelings while maintaining satisfactory signal quality (Supplementary Fig. 12). Figure 2b shows the change of pressure in the airbag during 5 pumping phases, where the pumping duration and pumping interval of the pump was set at 35 ms and 3500 ms for all five phases, respectively. The pressure increases from 0 to 12 kPa, corresponding to 0–90 mmHg, which is less than the systolic BP, and thus is suitable for continuous monitoring without discomfortable feeling generated (Supplementary Movie 1). Figure 2c shows the finite element analysis

(FEA) results and optical images of the airbag under different inner pressure, at 1 kPa and 10 kPa, respectively, from which regular deformations are generated with the silicone airbag, with maximum strain at 70% under Max. stress principle. It is worth mentioning that we developed a close-looped feedback strategy between the MCU and the mobile GUI to optimize the pressure in the airbag by real time adjusting/controlling the pump for optimizing signal quality. Specifically, the gradually decreasing pressure in the micro airbag and body movement would introduce fluctuations and distortions to the measured pulse wave. When distortion is detected, the pump will inflate the micro airbag to improve the signal quality (Supplementary Fig. 13). Supplementary Fig. 14 illustrates the flow chart of the close-looped feedback strategy, wherein signal quality assessment is performed by a peak detection algorithm (Supplementary Note 4) to determine if an effective piezo response was detected and then send instructions to MCU. The build-in counter in the MCU counts the pumping phases and inflating the airbag, allowing for a maximum of 5 pumping phases corresponding to a peak pressure of 12 kPa (Fig. 2b) before sending warning information for relocating to users via the mobile GUI.

With the assistance of the active pressure adaptation system for providing sufficient backpressure to the sensors, the piezo response generated from blood propagation can be effectively detected under wrist/elbow bending at different angles. Figure 2d shows the optical images of the sensor array mounted on wrist joint and elbow joint with a series of bending deformation, ranging from 0° to 45°. The measured piezo responses generated by blood propagation in radial artery and brachial artery are shown in Fig. 2e and Supplementary Fig. 16, where regular and stable response signal is detected under different deformations, with only slight increase in amplitude under 30° and 45° bending deformations. A continuous BP waveform typically contains three gradually attenuated peaks, which is caused by the interaction between the forward wave toward periphery generated by heart and the backward reflected wave from periphery (Fig. 2g)[42]. Consequently, the physical features of the BP waveform somehow can reflect the elastic property of the arterial tree, as well as the pressure propagation in the arterial tree. For instance, an early reflected wave may be caused by higher pulse wave velocity, which may cause the reflected wave reaching the central circulation in late systole rather than diastole[42,43], increasing the systolic BP. Hence, it is of crucial importance to extract features from the measured blood pulse wave for the accurate estimation of BP. The sampled piezo response (Fig. 2e), with a strong positive peak followed by a negative reverse valley and two weak peaks, is totally different from the BP pulse wave, with three gradually attenuated peaks. Therefore, we adopted a mathematical model to convert the piezo response to the original pulse waveform so that the physical features and time information can be accurately extracted for an accurate BP estimation (Supplementary Note 2).

The electromechanical coupling behavior of the piezo sensor is governed by the constitutive equations of piezoelectric materials (Eqs. (1)–(2), where the piezo response is regarded as the output of the system with the dynamic loading force generated from blood propagation (Fig. 2f). By applying inverse Laplace transform to the transfer function of the system, the dynamic output voltage can be quantitatively calculated with three parts (Eq. (3), Supplementary Note 2), including the integration of the output voltage, output voltage V and a constant C, where the coefficients A and B are related to the structural parameters and electrical parameters of coupling arrays. It has been reported that the integral portion dominates the dynamic output when the thickness of the piezoelectric materials exceeds several micrometers[41]. Therefore, we ignored the last portions in Eq. (3) and took the integral portion of output voltage for the conversion of dynamic loading force that is blood propagation wave (Fig. 2f). The measured piezo responses were converted to continuous blood propagation wave based on the mathematical model, as presented in Fig. 2g, where three gradually attenuated peaks can be clearly seen

regardless of in brachial artery or radial artery.

$$\sigma_{ij} = c_{kl}\varepsilon_{ij} - (e_{ik})^T E_j \quad (1)$$

$$D_i = e_{ik}\varepsilon_k + k_{ij}E_j \quad (2)$$

$$F(t) = A \cdot \int_0^t V(\tau)d\tau - B \cdot V(t) + C \quad (3)$$

In addition, blood propagation speed is also an important parameter that is highly related to the elasticity of the blood vessels, and thus can serve as a key determinant for BP[27,44]. PWV has been widely used[45,46] to improve the measurement accuracy of BP. However, global PWV reflects the elastic properties of arteries over a long segment, which might mask the initial variations in biomechanical properties in a small segment[47,48]. Moreover, incorrect estimation of blood travel distance would cause accuracy issue. In comparison, local PWV calculating pulse wave velocity in a short distance along arteries provides important diagnostic information of biomechanical properties for local artery walls[49]. However, the high propagation speed, normally ranging from 2 m/s to 20 m/s[50–52], as well as the short propagation distance between two adjacent sensing units bring an issue of short time difference (Δt) between two sensors' responses, and therefore requiring a high sampling rate[53] (>1500 data points/s) to reduce the error in Δt calculation, which arises a big challenge for wearable wireless transmission. To solve this problem, we develop a strategy associated with a resampling and transmission method to resample and wirelessly transmit the measured blood propagation waveform while maintaining the high sampling rate of the ADC (Supplementary Fig. 14). In this strategy, the piezo responses in the sensor array are sampled by the ADC with a high sampling rate at 4000 samples per second (SPS, 2000 SPS for each sensing unit), and then calculated in the MCU to extract Δt before resampling with a low frequency at 200 Hz (Supplementary Fig. 17). Figure 2h schematically presents the proximal and distal pulse wave, where a decreased amplitude in distal wave presents due to damping and biological absorption along the artery tree. The resampled piezo response and time difference is then wirelessly transmitted to the data model for BP estimation. Figure 2i presents the blood propagation waveform and local PWV measured on radial artery before and immediately after exercise, where the systolic peak and local PWV increase significantly after exercise compared to those during resting due to the stronger ventricular systole required to get more substantial cardiac output[54]. Besides, a stronger diastolic peak is observed after exercise due to high propagation speed (Supplementary Fig. 18).

## BP estimation model and performance evaluation

It is possible to get BP values from the raw pulse waveform, but respiration and body movements could lead to distortion of the signal waveform, thus affecting the accuracy[55,56]. Therefore, we used a third-order IIR bandpass filter to denoise the signal with a cutoff frequency of 0.5–8 Hz, which has been widely adopted to remove baseline drift and high frequency noise[57,58]. Before denoising, we resampled the signal to 1000 Hz for better signal restoration. Then we performed the integration calculation with the denoised signal (Supplementary Fig. 19). It was segmented into beat-to-beat waveforms for feature extraction with normalized amplitude (Window size of 1500 points with a 30% overlap). Many works have been proposed with PTT-based single indicator methods to estimate BP[59–61], but they could not satisfy the relevant standards[62,63] in sample size and require extra calibrations owing to insufficient features for model learning. Therefore, we proposed a multiple-feature fusion framework to construct the estimation model. It is meaningful to extract relevant morphological features, such as time information between different featured peaks/valleys and

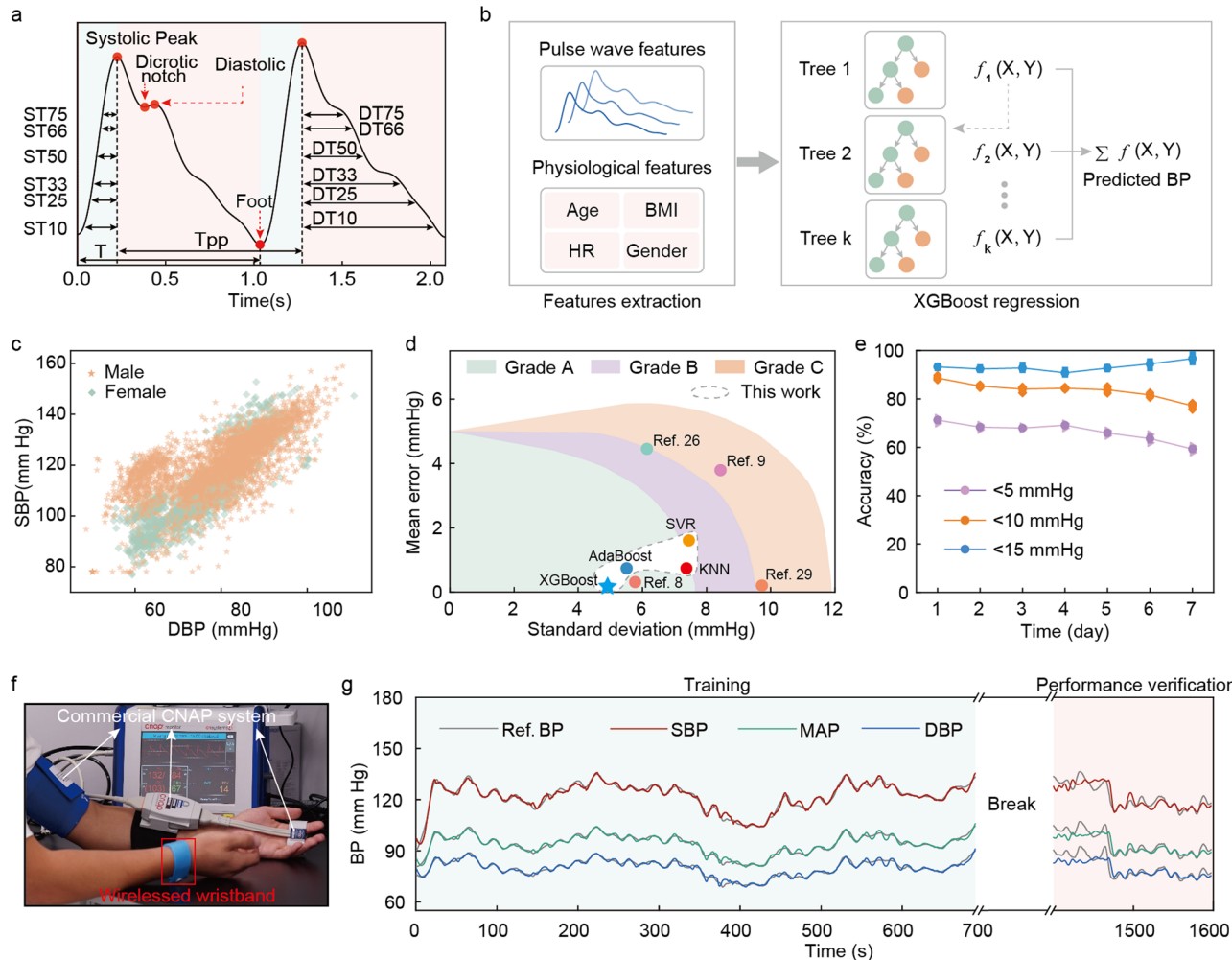

**Fig. 3 | Performance evaluation of the data model for continuous BP estimation. a** Illustration of the extracted pulse wave features, including systolic time span (ST) and diastolic time span (DT) and physiological features (**b**) in the estimation model. **c** Scatter plot of systolic blood pressure (SBP) and diastolic blood pressure (DBP) from 10 volunteers with wide dynamic BP range, which is necessary for accurate estimation of BP in a wide range. **d** Performance comparison in SBP of different estimation algorithms, including Support Vector Regression (SVR), Adaptive Boosting (AdaBoost), K-Nearest Neighbors (KNN), eXtreme Gradient

Boosting (XGBoost) utilized in this work (oval with white background) and other works, PPG[26], Bio-Z[8,9], Resistive pressure sensor[29], under the IEEE standard for wearable BP monitoring devices. **e** Model performance evaluation of SBP in a week after calibration ($n = 3$ tests on the same volunteer; center, mean; error bars, s.d.). **f** Optical images presenting the calibration process, with finger cuff of BioPAC system on the left hand, and wirelessed wristband on the right wrist joint. **g** Training and validation process showing a over 10 min trainning process, followed by 10 min and validation process.

relative amplitude since the pulse wave patterns are correlated with arterial blood pressure. Inspired by PPG-based blood flow analysis methodology[64], we extracted multiple morphological characteristics in the pulse waveform (Fig. 3a), four physiological features, and local PWV as the input to the training model (Fig. 3b). To effectively capture cardiac cycle characteristics from a single waveform, we located the following points in the segments: systolic peak, dicrotic notch, diastolic peak, and foot point (Supplementary Table 2). Time-domain characteristics (the x index of the extracted feature points) and morphological features (features extracted from waveform morphology and the features derived from the waveform) such as Systolic peak height, Diastolic peak height, Relative augmentation index and Inflection point area ratio. We constructed the feature vectors with morphological features, heart rates, ages, gender, and BMI as the input for the regression model. The SBP and DBP were extracted from segmented waveforms of continuous BP signals and used as reference labels for model training.

Before model development, a balanced data distribution and adequate sample size are important to minimize overfitting risk for model training. 17 volunteers covering a wide range of age and BMI

were selected to study the BP variations during the continuous BP measurement (detail process can be found in method). Besides, creating a wide range of blood pressure fluctuations can help to improve the model prediction performance for high blood pressure subjects. Hence, four subjects were asked to perform sustained hand grip and cold pressor (HGCP, Supplementary Note 6) during the tests to temporarily push the blood flow to increase BP, in which the subject squeezes the grip for 2 min to slowly elevate his BP gradually before immersing his hand into a bucket of ice-cold water (4 °C) for 1 min to further raise BP. Afterwards the subjects rested for 5 min to allow BP drops to near baseline. The measurement cycle was repeated at least three times and the visualized BP distribution is presented in Fig. 3c, where a wide range of BP values, from 48.6 mmHg to 158.87 mmHg, was achieved. Moreover, the mean SBP and DBP for male were 120.68 mmHg and 79.58 mmHg, respectively, and the corresponding values for female were 108.27 mmHg and 74.75 mmHg, respectively. The traces of sequential BP data of one subject were performed with model training process (Fig. 3f), and after a 30 min break, the data obtained from same measurement routine was used for BP validation (Fig. 3g).

In addition to data acquisition, model selection plays an important role for predicting results. Our data was collected from the participants who used our devices, which differs from the large-scale public databases that are often derived from controlled environments such as hospitals. Therefore, our model is suitable for the users in daily life scenarios. Considering possible overfitting risk of large models such as deep neural networks, a simple model may achieve better performance. Ensemble learning is a suitable candidate that can reduce the risk of overfitting. Extreme Gradient Boosting (XGBoost, Fig. 3b) is an extension of the Gradient Boosting Trees algorithm that combines the predictions made by multiple simpler models to produce a final strong model. It performs better than a single learner and takes less computational time than a complex network such as deep neural network. Thus, it was selected as the base model for BP estimation. To obtain the best model, we also tested the following machine learning models, including K-Nearest Neighbor (KNN), Support Vector Regression (SVR), and Adaptive Boosting (Adaboost) algorithm. All the collected data was preprocessed and separated the training and test sets in a 4:1 ratio with data shuffling before applying a fivefold cross-validation for the reliability of the results. The testing results show the XGBoost model performed the highest accuracy among all the tested models, with an accuracy of $-0.05 \pm 4.61$ mmHg for SBP and $0.11 \pm 3.68$ mmHg for DBP, which reached a Grade A according to the British Hypertension Society (BHS) standard[63] (Fig. 3d, Supplementary Fig. 20), significantly higher than that without PWV involved (Supplementary Fig. 21). We compared the estimation values with calibrated measurements, and the correlation analysis illustrated a high correlation between estimations and reference blood pressure, corresponding to the Pearson' r of 0.946 and 0.952 for reference SBP and DBP, respectively. As the Bland Altman plot shows, the data within the 95% confidence of interval indicates good agreement between two methods (Supplementary Fig. 22). To further improve the predictive performance and generalization ability of the model, we applied the Bayesian Optimization method to obtain the optimized hyperparameters. The evaluation metric is to obtain the minimum root mean square error (RMSE) value. The hyperparameters used in the model consist of maximum tree depth, the number of trees, regulation parameter gamma, L1 regularization term alpha, and learning rate, corresponding to 13, 437, 3.265, 0.006, and 0.039, respectively (Supplementary Table 4, Supplementary Figs. 23, 24).

To study the improvement of the active pressure adaption module on measurement accuracy and system robustness, the influence of pressure changes in the micro airbag on measurement accuracy was studied. Continuous BP wave comparison was conducted between the micro airbag just pumped and 30 min after the inflation and the micro airbag inflated with different pressure, ranging from 0 to 5 pumping phases, to study the influence of the changes in backpressure on measurement accuracy, as shown in Supplementary Figs. 25, 26, where SBP and DBP show small fluctuations with average BP fluctuates no more than 2 mmHg. Supplementary Fig. 27 shows the comparison in measured pulse wave with and without backpressure applied under the situations of wrist joint rotation, forearm raising and forearm shaking, where it can be found obvious signal distortion happened for the device without backpressure. In comparison, there is only slightly fluctuation in signal amplitude to a set of body movement, and no obvious signal distortion when backpressure is applied. It can be concluded that the introduction of the active pressure adaption module significantly improves the system robustness against body movement while maintaining measurement accuracy. Moreover, to demonstrate the robustness of our model to perform long-term estimation, we performed the data collection routine on one subject on day 1 and trained the model to predict BP results. The test routine was repeated 3 times per day for 1 week on the same subject. The pre-trained model predicted results for SBP can be found in Fig. 3e, where the trend of DBP changes was similar to that of SBP, with the mean

daily accuracy higher than that of SBP (Supplementary Fig. 28). It appears that the estimation results reached acceptable accuracy for 7 days. On day 7, the estimation accuracy of SBP still maintained at 59.29%, 77.15%, and 96.69%, corresponding to measurement differences <5 mmHg, 10 mmHg and 15 mmHg, respectively.

## Statistical and dynamic evaluation of the TSMS for wireless continuous and dynamic BP monitoring

To validate the measurement robustness of the TSMS, dynamic evaluation process was conducted. The evaluation is associated with wearing TSMS on the right wrist to continuously measure BP, and at the meantime commercial equipment (BioPAC) equipped with CNAP monitor, which has been proved providing clinically acceptable accuracy[65], attached on the same volunteers to measure reference BP, as illustrated in Fig. 4a and Supplementary Movie 2. Figure 4b shows the comparison between BP patterns measured by the TSMS and the CNAP of a volunteer, in which up to 25 min of continuous BP data is measured by the TSMS and agrees very well with those measured by CNAP, indicating the excellent performance of the TSMS. Besides, the statistical comparison between the TSMS and CNAP also shows excellent agreement, especially in DBP, with the same mean value as well as a slightly smaller dynamic range in 25 min (Fig. 4c, Supplementary Fig. 29). To further study the capability of our TSMS in dynamic monitoring of BP, hand grip and cold pressor process (HGCP) cycles, which has been proved an efficient way to increase BP due to the triggered sympathetic response[8,66], was adopted to realize a large dynamic range of BP (Fig. 4d). During HGCP maneuvers, the volunteer exercised with the hand grip for 2 min to slowly elevate his BP gradually before immersing his hand into a bucket of ice-cold water (4 °C) for 1 min to further raise BP. Then, a resting period for 5 min allows BP to drop to a low level. Bicycle HGCP maneuvers were conducted to raise a maximum SBP and DBP of 163 mmHg and 116 mmHg, respectively (Fig. 4e, gray dotted line). The continuous BP patterns measured by the TSMS show satisfying agreement, with the predicted maximum SBP and DBP at 158 mmHg and 109 mmHg, respectively. Figure 4f shows the statistical comparison between the measured BP and reference BP, which further proves the excellent agreement between the measured BP and reference BP, with negligible error in mean value of SBP and DBP although a slightly smaller dynamic distribution (Supplementary Fig. 30). To further validate the measurement accuracy of the TSMS system, a total number of 87 volunteers were selected to continuously monitor their BP for 2 min with BP measured by commercial CNAP and commercial sphygmomanometer (Omron HEM-7156T) as references. Figure 4g shows the statistic error distribution of the 87 volunteers with CNAP as a reference, where most of error bars are <10 mmHg regardless of SBP or DBP. Moreover, quantitative statistics on SBP and DBP show that excellent measurement accuracy was achieved with the TSMS, with over 70% for error <5 mmHg and over 98% for error <15 mmHg (Supplementary Fig. 31).

Hypertension is a highly prevalent condition with numerous health risks that could be caused by various factors. Among those pathogenic factors, biological aging and body mass index (BMI) have been reported showing positive association with BP[67,68]. To study how the age and BMI affect BP and further validate the TSMS as a reliable continuous BP monitoring system, we selected 17 volunteers, with comprehensive age and BMI, to perform a pilot study to monitor their daily BP with TSMS and take BP measured with a commercial sphygmomanometer, the gold standard for noninvasive BP measurement, as a reference. The volunteers were grouped into 4 teams based on their age and 3 teams based on their BMIs, respectively. The BP in four age groups, teenager (<20), youth (20–35), middle-aged (35–50) and elderly (>50), is presented in Fig. 4h and Supplementary Fig. 32, where a slightly increasing trend can be found in BP with the increased age regardless of gender. While there is a borderline hypertension individual in the age group of youth, which indicates the multiple pathogenic factors of hypertension. Besides, a hypertension individual with

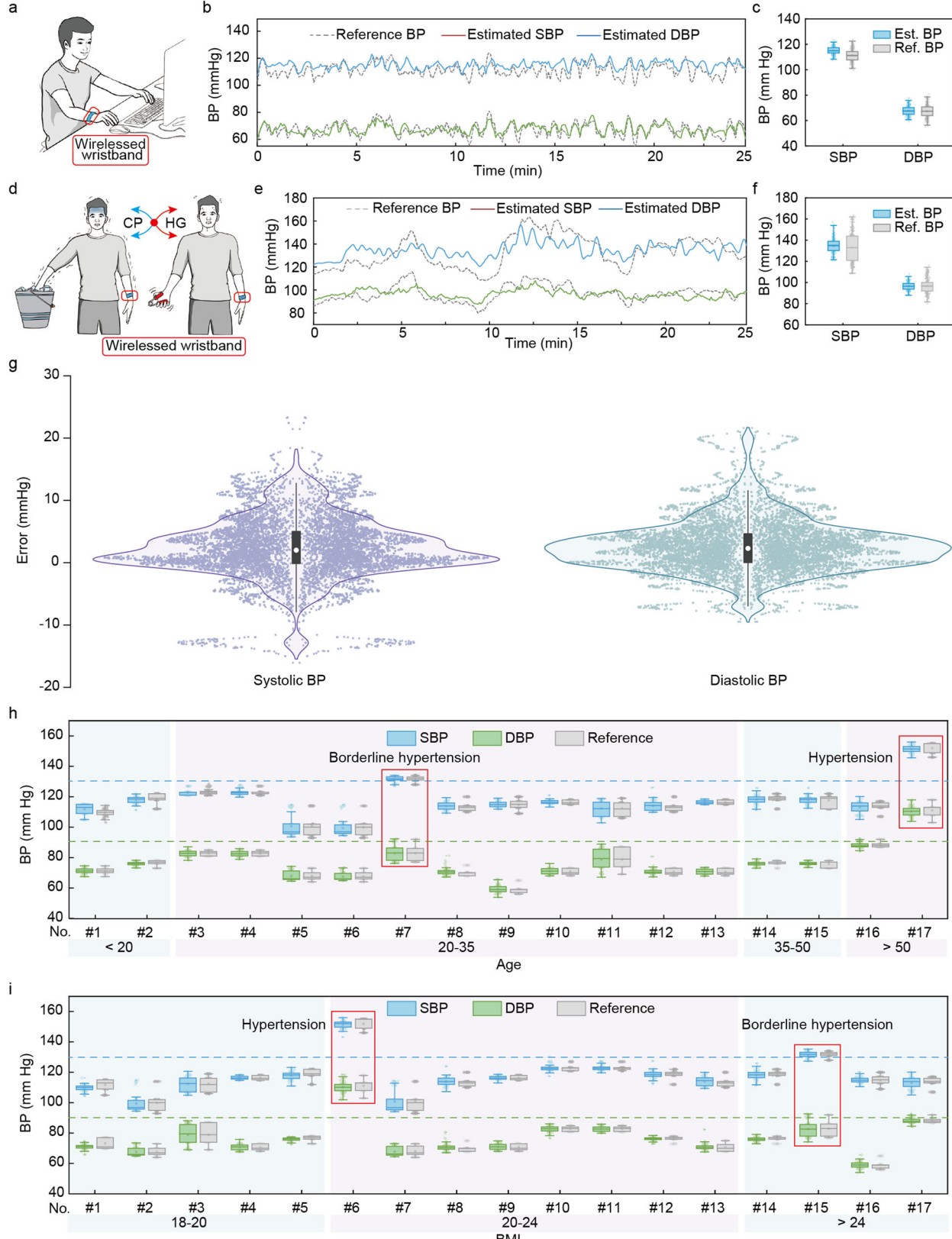

SBP over 150 mmHg and DBP over 100 mmHg was also recorded. To further study the correlation between BP and BMI and figure out the possible causative factors of the two hypertension individuals, the volunteers were grouped into slim (BMI < 20), normal (20 < BMI < 24) and overweight (BMI > 24), based on their BMIs. Figure 4i presents the statistical plots of the BP of the 17 volunteers, where the borderline hypertension individual locates in the overweight group, indicating the hypertension could be associated with high levels of BMI. Nevertheless, the high BP of the borderline hypertension individual could be caused by other factors since the rest of volunteers in the overweight group show normal BP. Besides, there is no obvious correlation between BP and BMI observed due to the limited sample size.

**Fig. 4 | Performance evaluation of the wireless wristband for continuous BP monitoring. a** Schematic illustration of continuous blood pressure (BP) monitoring in an office scene with the wireless wristband. **b** Performance validation of estimated BP (Est. BP), including systolic blood pressure (SBP) and diastolic blood pressure (DBP), in comparison with reference BP (ref. BP) measured by commercial continuous noninvasive artery pressure (CNAP) monitoring system, for 25 min. **c** Statistical plots present the comparison between our wireless wristband and BioPAC system in SBP and DBP. **d** Schematic illustration of hand grip and cold pressor process (HGCP) process for producing a wide range of dynamic BP. Volunteer's hand is immersed in cold water (4 °C) for 2 min, followed by hand grip exercise for 2 min to increase BP. **e** Predicted SBP and DBP in comparison with Bio-PAC during three HGCP cycles. **f** Statistical plots present the comparison between

our wireless wristband and CNAP system in SBP and DBP during HGCP cycles. **g** Violin plots representing the BP measurement accuracy of the TSMS compared with commercial CNAP in a total number of 87 volunteers ($n = 87$ volunteers with 39 male and 48 female involved. Circle point, median; central box limits, upper and lower quartiles; whiskers, upper and lower adjacent values, equal to 1.5 × interquartile range; outline, density plots with width equals to frequency; points, data points). **h** and **i** Statistical comparison of BP measured by the wireless wristband and measured by commercial cuff-based monitor in a total number of 17 volunteers, divided into 4 age groups (**h**) and 3 BMI groups (**i**). $n = 250$ BP data points in box plots of (**c** and **f**), $n = 80$ BP data points in the box plots of (**h** and **i**). Square, mean; center line, median; box limits, upper and lower quartiles; whiskers, 1.5 × interquartile range; points, data points, in box plots of (**c**, **f**, **h**, and **i**).

To further explore the significance of continuous BP monitoring by the TSMS, we tracked the continuous BP data of the borderline hypertension individual and hypertension individual for a week (Supplementary Fig. 33). Supplementary Fig. 33a presents the statistical BP variation of the hypertension individual in a week, where the BP maintains at a relatively stable value at the beginning while elevates over 10 mmHg before dropping to baseline in the next few days (-160 mmHg). Similarly, the continuous BP patterns for 5 min each day (Supplementary Fig. 33b, Supplementary Fig. 34) show the same trend while no obvious fluctuation during each day. Furthermore, the statistic BP data and continuous BP patterns for 15 min during a day is recorded in Supplementary Fig. 33c, d, from which the BP shows a higher value at midday, with SBP and DBP at 170 mmHg and 120 mmHg, respectively, and drops to a lower value at the night, with SBP and DBP at 158 mmHg and 108 mmHg, respectively. Similarly, continuous monitoring of the borderline hypertension individual (Supplementary Fig. 33 (e–h), Supplementary Fig. 35) shows that the BP drops to a normal range in the middle of the week, with minimum SBP and DBP at 119 mmHg and 85 mmHg, respectively, while raise to a maximum value at the end of the week, with the maximum SBP and DBP at 80 mmHg and 140 mmHg, respectively. Besides, it is interesting that the SBP fluctuated largely during a week while DBP, in turn, shows slight fluctuations. Meanwhile, the BP monitoring during a day shows that the BP rises to the maximum value at noon while the minimum value in the morning, which is similar to that of the hypertension individual.

## Discussion

Continuous BP monitoring is of vital importance for patients with chronic hypertension and can provide pre-diagnosis for initial patients. One typical application of the TSMS would be providing medical instructions for patients with sustained hypertension. Sustained hypertension patients with long-term intaking antihypertensive medication may appear normal BP when measuring in hospital. However, the antihypertensive drug might not control the BP effectively throughout the entire day and night, during which heart and kidney damage might be caused by the sharp BP fluctuation. In this case, continuous monitoring of BP at home can provide efficient advice for doctors to adjust the timing and dosage of the medication. Besides, patients with masked hypertension or borderline hypertension, whose BP is unstable and shows a wide dynamic range, may be misdiagnosed due to a normal clinical BP presented. Therefore, continuous BP monitoring plays a crucial role in developing diagnostic and therapeutic strategies for patients with masked or borderline hypertension. However, current wearable BP monitoring systems suffer from either bulky signal processing systems (ultrasound-based and bioimpedance-based) or cumbersome recalibration processes (ultrasound based and tonometry based), both of which have greatly hampered their uses in totally wearable/portable formats. Moreover, the unreliable interfacial contact between the device and artery will introduce extra noise signals to pulse wave, such as sliding noise caused by loose contact and occasional error, which will greatly reduce signal quality, and thus leading to poor measurement accuracy.

By integrating a sensor array, an active pressure adaptation system, and a signal sampling and processing module into a thin and soft wristband, we have demonstrated a powerful universal strategy that can provide real-time and reliable monitoring of continuous BP patterns. Excellent measurement robustness and accuracy, with $0.11 \pm 3.68$ mmHg for DBP and $-0.05 \pm 4.61$ mmHg for SBP, was achieved by the TSMS. All these capabilities rely critically on three main advances over previously reported works: (i) a flexible, self-powered piezoelectric sensor array that generates a reliable piezo response in response to blood propagation; (ii) an active pressure adaptation system associating with backpressure adjusting and a multiple pumping control strategy to guarantee good contact between the sensor array and the skin; (iii) a machine learning based data model with a wireless communication strategy for local PWV transmission and continuous BP estimation. Benchtop studies, structural design, theoretical simulation, and initial trials on a total number of 87 volunteers have demonstrated the feasibility and practical utility of the TSMS in continuous BP monitoring. Furthermore, dynamic BP tracking of a hypertension individual and a borderline hypertension individual reveals the BP changes during a day and a week. We envision that the TSMS could play a crucial role in realizing precise BP control for hypertension individuals and cardiovascular disease prevention through continuous BP monitoring and the development of personalized diagnosis and treatment schemes. To realize long-term accuracy monitoring of continuous BP pattern, comprehensive pre-train process to the data model is needed with a balanced distribution on BP range and other factors that contribute to BP fluctuations. This study, instead, focused on limited size of human participants actual scenes. We believe further expanding the number of human participants and improving BP distribution will improve the BP estimation accuracy.

## Methods

### Fabrication of piezoelectric transducer array

The fabrication process (Supplementary Fig. 36) started with spin coating a thin layer of polydimethylsiloxane (PDMS, Sylgard 184 silicone elastomer, with the cross-linker ratio at 10:1), with the thickness of 200 μm on the glass substrate (75 × 75 mm) to serve as the adhesion layer. Then a layer of polyimide (PI) film (with the thickness of 25 μm) was laminated to the PDMS with a roller to remove air bubbles. The PI film is then patterned with a high precise ultraviolet (UV) laser cutter (ProtoLaser U4, LPKF), with the laser wavelength at 355 nm, the maximum laser power at 5.7 W and the pulse frequency ranging from 25 kHz to 300 kHz, to serve as the supporting layer. Then the patterned PI traces was cleaned with acetone, ethanol, and deionized (DI) water sequentially before depositing the thin film metallic electrode layer (Au/Cr with a thickness of 200 nm/10 nm) with an electron-beam evaporation system (E-Beam, EBS-500F, Junsun Tech). Removing the PI layer enables the formation of the electrode traces (Supplementary Fig. 3). Water-soluble tape was utilized to pick up the electrode traces from the PDMS substrate before depositing the adhesion layer Ti/SiO$_2$ (5 nm/50 nm) on the PI side by E-Beam. The electrode traces together with the PDMS substrate (200 μm in thickness) were exposed to UV-

induced ozone for 5 min before heating in the oven at 80 °C for 10 min to enable the formation of covalent bonding between $SiO_2$ and PDMS. Removing water-soluble tape and welding PZT to the electrodes with silver paste at 100 °C for 10 min enable the robust connection. Finally, injection of PDMS with a syringe, following by degas and curing at 80 °C for 1 h facilitated the removal of interfacial gaps and encapsulation of the sensor array.

### Fabrication process of the one-way valve

Firstly, the silicone hose was fabricated by casting silicone elastomer (Ecoflex 00-30) in a 3D printed mold, with the inner diameter of 2.8 mm, outer diameter of 4.7 mm and 4.2 mm in thickness, following by degassing, curing at 80 °C for 10 min and punching hole (0.5 mm in diameter) sequentially (Supplementary Fig. 9). Then a thin layer of silicone elastomer (Ecoflex 00-30), with the thickness of 200 μm, was spin-coated on a glass sheet before being cut into circle (4.7 mm in diameter) by a UV laser cutter (ProtoLaser U4, LPKF). Cleaning the substrate with isopropanol (IPA), ethanol and DI water sequentially and peeling off the silicone film from the glass sheet enables the fabrication of silicone cover. Subsequently, the isolation PI film (25 μm in thickness) was fabricated by UV laser cutter. Finally, the silicone hose, isolation film and silicone were selectively glued on the predefined bonding sites.

### Fabrication process of the silicone airbag

The first part of the micro airbag was fabricated by casting silicone elastomer (Ecoflex 00-30) with the assistance of a 3D printed mold (Supplementary Fig. 4). The first part of the micro airbag was cleaned with DI water and blow-drying by nitrogen before bonding to the silicone cover (200 μm in thickness) at 80 °C for 1 h to enable the stable covalent interfacial connection. Finally, the outlet was connected to commercial silicone hose (1 mm in outer diameter, 0.5 mm in inner diameter) and sealed by silicone glue for ease connection with the one-way valve and micro pump.

### Volunteer data acquisition

In this work, seventeen volunteers with different sex and ages (9–62) participated in the experiments. The participants were required to stay in a sitting posture, with the cuff around the left upper arm and the sensor on the fingers, and the other arm wore our pulse sensor on the wristband (Fig. 3g). To synchronize our pulse signal with BP signal, we additionally measured the PPG signal on the finger by PPG sensor (BIOPAC Systems Inc.) for signal correlation. The subjects were allowed to rest for 5 min before the data acquisition process. Before the blood pressure measurement, the calibration process was performed automatically by the device to ensure accuracy. The experiments were conducted using a medical continuous BP monitoring device and computer-based data acquisition system (NIBP100D, BIO-PAC Systems Inc.). The subjects' blood pressure and pulse signals were recorded simultaneously with a sampling rate of 2000 Hz. Our experiments were conducted with few arm movements to avoid noticeable motion artifacts. The duration of each experiment was over 30 min. Participants' overall BP distribution is provided in Fig. 3d, the values of which were 120.68 ± 11.86 mmHg (MD ± SD) of SBP and 79.58 ± 11.36 mmHg of DBP for males, and 108.27 ± 12.06 mmHg, 74.75 ± 8.12 mmHg for females, respectively. All human experiments were conducted by protocols approved by the Human Subjects Ethics Sub-Committee of Research Committee, City University of Hong Kong, Hong Kong, China, and guidelines were followed.

### Flexible printed circuit board and communication

To make the device wearable and easy to use, a thin flexible printed circuit board (FPCB) was designed. Two photolithographically patterned copper conductor layers (17.5 μm thick) on polyimide substrate are used to connect the circuit, with an overall FPCB thickness of only 139 μm. The electronic components are soldered to the FPCB using solder paste and heat gun to fabricate the control unit. This circuit can collect the voltage data generated by the sensor in a relative high speed (like 4000 samples per second) and transmits the results to the phone via Bluetooth Low-Energy (BLE) wirelessly. At the same time, it can control the air pump to produce proper pressure on the sensor. To achieve these functions, a micro-controller unit (MCU) with integrated Bluetooth functionality (CC2640R2F, TI.), a 24-bit high speed analog-to-digital converter (ADC, ADS1258, TI.), a low dropout regulator (LDO, TPS76933, TI.), a multiple-output regulator (TPS65135, TI.), air pump driver chip (lp6832, Vito Fluid), 2.4 GHz impedance matched balun (2450BM14G0011, Johanson Technology), 2.4 GHz mini antenna (2450AT18A100E, Johanson Technology) and passive components such as resistors, capacitors and inductors are integrated into this circuit. The LDO and multiple-output regulator is used to convert the 3.7 V output voltage from the Li-ion battery to 3.3 V and ±1.5 V to power the ADC and MCU. The ADC can collect the small signal generated by the sensor in a relative high speed. After that, MCU can access those data by Serial Peripheral Interface (SPI) and sent them to the phone via Bluetooth after a simple processing (Supplementary Fig. 38). The mobile phone can monitor the operation status of the device in real time, and if the signal-to-noise ratio of the collected data is found to be low, it can send commands to start the air pump. After receiving the command, the MCU will control the air pump driver chip to drive the air pump to generate a proper pressure to improve the signal to noise ratio. All these functions are integrated in a $52 \times 27 \times 3.5$ mm control unit. Since the substrate of the circuit is FPCB, the whole circuit can be bent to fit the shape of the user's arm, making it more comfortable to wear (Supplementary Fig. 37).

### Characterization

The electric related data, current and voltage, were sampled by the multichannel Data acquisition system (DAQ 6510, Keithley) with a constant sampling frequency at 1000 Hz. The dynamic loading was applied with a universal testing system (5924, Instron), which allows precise control on loading force and loading speed. The photographs of morphology changing of the micro airbag were taken by a high-speed camera (OSG130-120UC, Yvsion) with a recording frame rate at 300 fps. Thermal images were taken by fixed infrared thermal camera (FLIR A325sc).

### Ethics statement

All procedures during the olfaction system testing from human participants are approved by Human and Artefacts Ethics Sub-Committee, City University of Hong Kong Research Committee with the reference number of HU-STA-00000393. The informed consent of all participants was obtained prior inclusion in this study.

### Reporting summary

Further information on research design is available in the Nature Portfolio Reporting Summary linked to this article.

## Data availability

The data that support the findings of this study are available from the corresponding authors upon request.

## Code availability

The data that support the findings of this study are available from the corresponding authors upon request.

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

## Acknowledgements

This work was supported by InnoHK Project on Project 2.2—AI-based 3D ultrasound imaging algorithm at Hong Kong Centre for Cerebro-Cardiovascular Health Engineering (COCHE), City University of Hong Kong (Grants No. 9667221, 924007 and 9680322), and National Natural Science Foundation of China (Grants No. 62122002).

## Author contributions

J.L., H.J., J.Z., and X.H. contributed equally to this work. J.L, X.Y. and X. M. conceived the ideas and designed the experiments. J.L, H.J., J.Z., S.J., X.T., R.H.C., Y.Z., and X.Y. wrote and revised the paper. J.L, H.J J.Z., X.H., Z.G., K.Y, D.L., B.Z., Y.L., Ya.H, Y.H, G.Z, Z.X., JY.L., C.Y., Y.G., M.W., Y.J., and Q.Z. performed experiments and analyzed the experimental data. J.L., and L.X., performed structural designs and simulations. H.J., and J.Z performed the data model, circuit design and software programming.

## Competing interests

The authors declare no competing interests.
