## [Peer Review file · Nature Communications]

REVIEWER COMMENTS

Reviewer #1 (Remarks to the Author):

In this work, the author presents a wearable system for continuous wireless monitoring of artery blood pressure. The author implemented a wireless monitoring system for artery blood pressure. However, this work lacks novelty compared to previous reports on the bulk piezoelectric blood pressure (BP) sensor ("Adv. Mater., 34, 2110291, 2022") and BP estimation model ("IEEE Sensors Journal, 21,11,12498-12510, 2021") exist. Additionally, due to a small number of sample measurement and sensor structure, it seems that BP sensor lacks the reliability including measurement distortion. The reviewer cannot recommend this manuscript for publication in Nature Communications.

1. A study on converting the response into a BP waveform using bulky piezoelectric materials was published in Adv. Mater., 34, 2110291, 2022. Compared to previous papers, there is no novelty in terms of material or BP waveform calculation process, and only the BP estimation process was changed, followed by post-processing

2. Papers such as "IEEE Sensors Journal, 21,11,12498-12510, 2021," and "IEEE Sensors Journal, 22, 3, 2475-2483, 2022," have measured BP by machine learning based on PWV values and multiple physiological features. There is no significant difference in the BP calculation method compared to previous papers. This system does not show any improvement like higher accuracy in BP estimation and sensor sensitivity.

3. Does measuring the pulse wave velocity (PWV) at a short distance have medically significant results? Conventional PWV is measured between two arteries that are far apart. Therefore, PWV values at distant points are considered to better reflect cardiovascular parameters. Further explanation is necessary to show that the local PWV reflects cardiovascular parameters medically.

4. In the text, it was expressed that "grade A according to the British Hypertension Society." The criteria for Grade A are shown in the table only as average and standard deviation values. Additionally, there is no standard for the total number of subjects to be measured. The sample size of 17 subjects is insufficient to demonstrate the reliability of the measurement. International standards require a minimum of 85 subjects using an auscultatory sphygmomanometer as a reference.

5. A micro-airbag is included with the sensor to overcome the conformal attachment to the skin due to mechanical stiffness from bulky piezoelectric materials. However, there may be distortion in the measured waveform due to the pressure applied by the micro-airbag. In the text, it was expressed that the pressure of 0-12 kPa (0-90 mmHg) with a micro-airbag is less than the systolic BP value, making it suitable for continuous monitoring without discomfort. However, even if the systolic BP is over 100 mmHg, the pressure felt on the skin is much smaller than this, so the effect of the pressure from the micro-airbag is greater than the actual arterial movement on the skin. Therefore, data showing no distortion of the BP waveform for various micro-airbag pressures should be provided."

6. Further revisions are required to enhance the linguistic and grammatical precision of the entire manuscript, with particular emphasis on achieving consistent adherence to the conventions pertaining to units and spacing.

Reviewer #2 (Remarks to the Author):

This work reported a continuous blood pressure monitoring strategy with the ability of active pressure adaption and in-situ transformation of sensory signals. Such a thin, soft, miniaturized system can achieve a high measurement accuracy of 0.11 ± 3.68 mmHg DBP and -0.05 ± 4.61 mmHg SBP, which has reached the Grade A level of standard wearable BP monitoring devices. This work will definitely be a remarkable promotion into this field and will attract broad attention in the field of CVDs. To improve this manuscript, some personal suggestions are provided for further discussion and some minor concerns should be addressed before the publication of this paper.

1) L52. In the introduction, to my knowledge, the mean blood pressure may be a calculated value based on the formula $MAP = [SBP + (2 \times DBP)]/3$ and not a recording index in clinic.

2) Grade A should be provided with one reference or a detailed description for readers' better understanding. Which standard in the British Hypertension Society (BHS) is referred to for this evaluation?

3) For a better reading, supplementary table 1 is suggested to be further clarified. For example, some sensors are described in terms of the method. A consistent unit should be better for understanding the mentioned devices' dimensional differences. How to define dynamic and static BP and why is the continuous feature classified into static and dynamic BP? It is better to provide the type of commercial products if it is summarized in the table.

4) Regarding the airbag system, obtaining stable pressure in the airbags is critical for achieving high accuracy in BP monitoring in my opinion. Generally, maintaining such a stable pressure requires a closed-loop system, including an air pump, a pressure sensor, and valves. In this work, the pressure in the airbag is fluctuant as I understand. The authors should address why this design is preferred to some traditional closed-loop systems. The other question is about the motion artifact, which is the most critical challenge in continuous BP monitoring at present. How to consider it based on this design.

5) Extended Data Figure 5f, force(mV)? How about the sensitivity of the used piezoelectric pressure sensor?

6) About the "denoise the signal with a cutoff frequency of 0.5 to 8Hz", is it possible to provide a detailed discussion of why the cutoff frequency is 0.5 to 8 Hz?

7) The output voltage of the piezoelectric sensor seems to be not stable as the authors described. The authors should provide a detailed discussion or necessary evidence about what will be caused by this fluctuation in BP monitoring and whether it can be mediated by the loads applied by the airbags.

8) In the main section, the description of the signal processing module is vague, could you explain its functions?

9) L112-114. I prefer “with the top electrode connected to the bottom substrate via a vertical connecting point (VCP) for optimized mechanical robustness, and ease of electrical connection, enabling the sensors to adapt to various mechanical deformations” rather than “the top electrode is routed to the bottom substrate...”.

L117-121. The sentence is hard to read. I suggest rephrasing it “The pair of PZT sensors is encapsulated with soft silicone, providing soft and conformal contact between the sensors and skin, thus allowing for accurate conversion of local deformation of the sensor caused by the expansion/contraction of the artery into electrical outputs.”

L125-127. I prefer the below description: “By measuring the pulse travel time (PTT) of a single pulse waveform arriving at two sensors, and the distance between the two electrodes, the local Pulse Wave Velocity (PWV) can be calculated.”

10) L146. Please be more specific. If the measurement of “6 times measures per hour” is intermittent, and how long each measurement takes?

11) In Supplementary Fig. 5, the figure legend of Supplementary Fig. 5(d) doesn't match the data presented in the figure, please revise it.

12) Figure 2 h. The figure shows the pulse wave of two sensors displayed a different amplitude. Could you please explain the reason?

13) Could you please annotate the subscripts of the variables in the equations of the chapter “piezo response conversion” to facilitate understanding?

14) According to the description of L249: The signal was resampled to 200Hz, which does not match the frequency for preprocessing, please explain the reason.

15) Please provide more information about how the system detects if a signal is available for control of the micropump.

Reviewer #3 (Remarks to the Author):

Overall, the article is well-written and informative, providing insight into the importance of monitoring blood pressure and the development of wearable devices for continuous non-invasive blood pressure

monitoring. The integration of materials, devices, mechanics, data processing methods, and integration strategies for the system is informative, and the advantages of the system are clearly outlined. However, it is recommended that certain issues related to professional and grammatical be addressed prior to publication.

Professional aspects

Main

Line 91: Provide more detail on the methodology, it would be helpful to provide more detail on the methodology used to design and test the system. For example, what specific PZT materials were used, and how were they chosen?

Line 102: The main section ends with the potential benefits of continuous BP monitoring. However, it could benefit from a more explicit statement about the purpose or objective of the study, to give readers a better sense of what the article will be focused on.

Results:

Line 104: The article would benefit from a more structured approach. The authors could use headings and subheadings to organize the information and make it easier to follow in Design and working principle of the wireless integrated BP monitoring system section.

Line 675: Figure 1: The information provided in (g) about PPG and Ultrasound not sufficiently clear, please revise it.

Line 106, 150: Use consistent terminology: The text uses multiple terms to refer to the BP monitoring system, including "wearable system," "wireless system," and "flexible, lightweight wristband." Using consistent terminology can help readers follow the description more easily.

Supplementary Fig. 6. In Fig 6 (c), It appears that the system's response time is faster when the one-way valve is not assembled compared to when it is. Please explain.

Line 162: it could benefit from more specific details on the materials and components used. For instance, it would be helpful to know the specific type of PZT used for sensors, as well as the acoustics performance of the sensors.

Line 183-184, "the change of pressure in the airbag during 5 pumping phases, where the pumping duration of the pump was set at 35 ms for all five phases," it is unclear whether the five phases have the same interval or different intervals. A more detailed explanation of the pumping phases and their associated intervals would greatly enhance the understanding of the experimental setup.

Line 185: It is unclear what is meant by "less than the systolic BP" in the sentence "The pressure increases from 0 to 12 kPa, corresponding to 0-90 mmHg, which is less than the systolic BP," It would be helpful to explain this further or provide a reference.

Line 189-191: Consider adding some contextual information for the reader to better understand the significance of the results presented about the close-looped feedback strategy. For example, why do we need closed-loop control instead of open loop control here, is there any disturbance? If so, is the control strategy robust enough to against it? It would be helpful to explain this further or provide a reference.

Line 199: It would be useful to clarify what is meant by "different situations." Are the situations referring to different wrist elbow joint angles?

Line 214-215: it would be useful to clarify what is meant by "physical features" and "time information" that are extracted from the original pulse waveform.

Line 257-259: it would be helpful to provide a citation or reference for the claim that respiration and body movements could lead to distortion of the signal morphology.

Line 265: it is recommended to clarify what is meant by "relevant standards" and provide a reference or citation.

Line 281: Provide more information about the selection criteria for the volunteers to help readers understand the representativeness of the sample.

Line 296: Consider clarifying what is meant by "limited sample size". How many samples were included in the study, and why is this number considered to be limited? Providing this information will help the reader understand the significance of the study findings.

Line 297: Consider including a brief description of how the XGBoost algorithm works rather than just in Fig 3b. This will help readers who are not familiar with the algorithm to understand why it was selected as the base model.

Line 329: Provide more context for the comparison with BioPAC: The paragraph compares the BP patterns measured by the wireless wristband with those measured by BioPAC, but it doesn't explain what the accuracy of products BioPAC is or why it's being used as a reference.

Line 356: It would be helpful to explain that the cuff-based sphygmomanometer is considered the gold standard for BP measurement and was used as a reference to compare the accuracy of the wristband readings.

Discussion:

Line 394 and 404: Please use more precise language here. It contains some imprecise language that could be clarified to improve its professional tone. The phrase "may cause heart and kidney damage due to sharp fluctuation of BP" could be revised to more accurately reflect the relationship between fluctuating BP and organ damage. Similarly, the phrase "unreliable interfacial contact between the device and artery" could be revised to more specifically describe the issues with signal quality and measurement accuracy.

Line 424: Consider providing more information on the limitations of the study, such as the small sample size or the potential confounding variables. This context would help the reader understand the study's limitations and potential biases.

Grammatical aspects:

Main

Line 48: The sentence "..., heart rate, cardiac output, which are closely relate to CVDs" should be revised to "..., heart rate, and cardiac output, which are closely related to CVDs."

Line 97-100: The sentence "Compared to other cuffless BP monitoring devices, ..., cuffless blood pressure monitoring devices" should be divided into two sentences like: "Compared to other cuffless BP monitoring devices, the integrated system reported here shows great advantages in terms of wearability, continuance, dynamics, and measurement accuracy. It is characterized as Grade A based on the standard for wearable, cuffless blood pressure monitoring devices." The overall structure of the main could be improved by breaking down longer sentences into shorter ones and avoiding the use of passive voice wherever possible.

Results:

Line 156, In the sentence "To our best knowledge,...,wearability, accuracy, continuance, dynamics, ...," "continuance" seems like an unusual word choice. It could be revised to durability.

Line 162-164: The sentence "While the use of PZT as sensors based ... stiffness," it would be clearer to write "While the use of PZT as sensors is based on its excellent electrical, physical and chemical stabilities, it also leads to interfacial contact issues with the skin due to its mechanical stiffness."

Line 165 and 166: there are a few instances where plural and singular forms are mixed, such as "micro-airbag" and "micro pump" in the same sentence. It would be clearer to use consistent grammar throughout.

Line 194: 1. The sentence "The build-in counter in ... for relocating to users via the mobile GUI" has a grammatical error. It could be rephrased as "The built-in counter in the MCU counts the pumping phases and inflates the airbag, allowing for a maximum of 5 pumping phases corresponding to a peak pressure of 12 kPa (Fig. 2b) before sending warning information to users via the mobile GUI to relocate."

Line 248: Check for typo: "senor array...", should be "sensor array"

Responses to comments of Referee #1

Comments from Referee #1:

Summary Comment: In this work, the author presents a wearable system for continuous wireless monitoring of artery blood pressure. The author implemented a wireless monitoring system for artery blood pressure. However, this work lacks novelty compared to previous reports on the bulk piezoelectric blood pressure (BP) sensor (“Adv. Mater., 34, 2110291, 2022”) and BP estimation model (“IEEE Sensors Journal, 21,11,12498-12510, 2021”) exist. Additionally, due to a small number of sample measurement and sensor structure, it seems that BP sensor lacks the reliability including measurement distortion. The reviewer cannot recommend this manuscript for publication in Nature Communications.

Our response: We thank the referee for the time and comments. As the reviewer mentioned, there are some published papers related to either BP measurement devices (“Adv. Mater., 34, 2110291, 2022”) or BP estimation models (“IEEE Sensors Journal, 21,11,12498-12510, 2021”). However, these reported works are distinguished with ours in the aspects of the device design, system-level integration, weight, formats, and most importantly the BP estimation accuracy level, as the accuracy is the most important and remaining challenge to the wearable, user friendly and real time BP measurement devices. To elaborate these points, we give a detailed illustration on each paper the reviewer mentioned.

For the “Adv. Mater., 34, 2110291, 2022” paper, the authors used the continuous electric output (voltage) as the continuous pressure wave upon the artery and calibrated the continuous pressure wave with an intermittent commercial medical sphygmomanometer (U32K, Omron Ltd., Kyoto, Japan) to get continuous BP wave. It is worth noting that during the calibration process, normally *a minimum measurement duration of 30s* is required for the discrete sphygmomanometer to get a systolic BP and a diastolic BP, during which the fluctuation of the pressure waveform significantly influence the measurement accuracy. Besides, the authors regarded the continuous pressure wave as the continuous BP wave by using a calibration factor, which means parameters that contribute to the voltage output, such as the initial back pressure provided by the tape and tiny joint movements, will greatly affect the measured pressure wave, and thus the BP measurement accuracy. Most importantly, the calibration factor ϕ between the measured pressure wave and the BP wave differs from user to user, even two times measurement on the same user (the back pressure provided by the tape changes after the replacement of the sensor), which *raises the calibration requirement each time before measurement*. In comparison, our integrated system provides a totally wearable interface, with sensor array and signal processing module encapsulated in a portable silicone wristband. The active pressure adaption module provides a universal strategy for skin-integrated electronics that suffer from poor interfacial performance.

With the assistance of the model-driven data model, continuous BP can be effectively estimated without the need for repeated calibration process while maintaining satisfying accuracy for a period after the model training process (**Fig. 3e**).

For “IEEE Sensors Journal, 22, 3, 2475-2483, 2022” paper, two commercial rigid PPG sensors, 0.055m away from each other were adopted for the measurement of local PWV, with a sampling rate at 50 Hz, before transmitting the data to computer for BP estimation. For the local PWV measurement, over 5 m/s for radial artery, a high sampling rate is normally required (over 1.5 kHz) for a satisfied measurement accuracy due of the short propagation time of pulse wave between two measurement sites. Such a high sampling rate also brings high demands on the wireless communication technology, which of course will cause hurdles for wearable devices used in daily life instead of in laboratories. To solve this problem, we proposed the resampling strategy for the realization of high sampling rate (20 kHz) as well as wireless communication between devices and smart devices (smart phone or computer), as illustrated in **Extended Fig. 6b**.

In summary, compared with the above two papers, we presented a totally wearable, highly integrated and wireless devices for continuous BP monitoring. The active pressure adaption module and the signal processing strategy enable continuous and wireless measurement of BP while showing excellent measurement accuracy in various situations, both in continuous static monitoring (**Fig. 4a**) and continuous dynamic monitoring (**Fig. 4d**). We have added these relevant papers in **Supplementary Table 1**, and the comparison between our work and these relevant references are added in Line 46 in supplementary.

Modifications: In Line 46, supplementary, we added the comparison between published works (wearable sensors/BP estimation models) and our work. We modified the text as “Conventionally, non-invasive blood pressure is measured with a manual cuff, in which a pressure gauge is equipped to measure the air pressure inside the cuff. By inflating the manual cuff to a pressure that is higher than BP, systolic BP and diastolic BP can be recorded with the assistance of the stethoscope. While continuous BP measurements typically associate with invasive methods. Therefore, it’s extremely important to develop wearable devices for continuous BP monitoring. To date, there are mainly four categories of technologies that are developed for continuous BP monitoring (**Supplementary Table 1, Supplementary Fig. 1**): (1) Optical based technology, i.e., Photoplethysmography (PPG) measures the changes in reflected light caused by volumetric variation of blood circulation for the estimation of BP; (2) Acoustic based technology, i.e., Ultrasound wall-tracking derives the changes in artery diameter by analyzing the echo signals from anterior vessel wall and posterior vessel wall to calculate continuous BP; (3) Electrical based technology, i.e., Electrodes array measures the variation of bioimpedance generated by blood propagation to predict BP; (4) Pressure sensor based technology, i.e., High precise pressure sensor, as presented in this work, are adopted to detect dynamic pressure from artery generated by blood

propagation for the estimation of BP.

Our wireless integrated system employs a high precise, self-powered piezoelectric sensor array to detect the pressure variation of artery caused by blood propagation, providing more stable and power saving strategy compared to PPG. Besides, compared to acoustic and electrical methods, in which bulky signal sampling and processing equipment or complicated interfacial connection are required, our device provides a totally wearable interface for users, with all the components integrated into a wristband. The backpressure generation system in our devices provide powerful supporting for the sensor array to solve the common drawbacks, such as unstable interfacial contact and resulted poor signal quality in pressure sensor based technologies.

Additionally, from the point of BP estimation algorithms, there are several commonly used algorithms BP estimation.

(1). K-Nearest Neighbor (KNN): This algorithm calculates the distance between the input values and the k nearest sample points in the training set make predictions about the output of the input values. It can be used in nonlinear data but when it comes to large amount of data, the calculation cost is high.

(2). Support vector regression (SVR): The goal of this algorithm is to find an appropriate line to approximates the input variables and predict values. It can handle nonlinear and high-dimensional data by using kernel functions, but it might be sensitive to noise and the selected parameters needs to be tuned to optimize the model.

(3). Adaptive boosting regression (ABR): This is an ensemble learning algorithm that uses a combination of weak learners to make the prediction values. It can improve generalization, but it may be sensitive to noise and anomalous data.

(4). Multiple linear regression (MLR): This algorithm uses a linear equation to predict outcomes from independent variables. It is straightforward and easy to apply, but it may not fit nonlinear data well and it needs pre-assessment of the relationship between variables.

(5). Artificial neural network (ANN): This algorithm uses a network of multiple connected neurons to learn from input data and make predictions. It can capture nonlinearity in the features, but it risks overfitting and demands a lot of data and computation.

We applied XGBoost regression algorithm for BP estimation considering the fast computation speed and high accuracy and performance. Moreover, it does not require a large amount of data for training and can be easily optimized by fine-tuning various hyperparameters and avoiding the overfitting problems that often occur in large models such as deep neural networks.”

Supplementary Table 1. Technical comparison of continuous BP monitoring.

Method	Wireless? Totally wearable?	Continuous?	Accuracy				Device dimension
			DBP (mmHg)		SBP (mmHg)		
			ME	SD	ME	SD	
PPG ¹¹	Yes PPG sensor and hard sampling board in a watch shaped wrist.	Not	-0.07 ± 4.47		0.00 ± 3.61		Not reported.
PPG ⁸	Not Commercial PPG sensor and ECG electrode required.	Yes Beat to beat. Static BP.	3.23 ± 4.75		4.43 ± 6.09		Not reported.
PPG ¹²	Yes	Yes 1000s long. Continuous. Static BP.	Not reported.		0.24	1.18	Device size 4 cm × 2 cm Board size Not reported.
PPG ¹³	Yes Neither the sensor nor the electrical circuit are designed in wearable format.	Not	2.12	0.26	2.94	0.72	Not reported.
Piezoelectric ultrasound transducer ¹⁴	Not Connecting to sampling equipment required.	Yes 30s long. Continuous. Static BP.	Not reported.				23 mm × 20 mm
Resistive pressure sensor combined with ECG ¹⁰	Yes Wearable pulse sensor and ECG electrodes are worn separately.	Yes 4 hour long. 15s time window. Static BP.	0.24	5.19	0.07	9.66	ECG patch 7 cm × 2 cm Pulse sensor 4 cm × 1.7 cm
Piezoelectric pressure sensor ³	Not Sensor in wearable format. Rigid sampling board. Sensor and electrical board were not integrated.	Yes Static BP.	Not reported.		Not reported.		Device size 3.5 mm in diameter. Board size not reported.
Capacitive pressure sensor ¹⁵	Not Connecting to sampling equipment required.	Yes 10 min. long. Static BP	0.48 ± 1.96		1.43 ± 1.96		8 mm × 8 mm
Capacitive pressure sensor ¹⁶	Not Fixing wristband and sampling equipment required.	Yes 9 min. long. 70 beats window. Static BP	-0.054 ± 2.09				15 mm × 2 mm
Commercial silver electrodes. Bio- impedance ⁹	Not Bulky wristband and sampling board required.	Yes 60 min. long. 5-10s time window. Dynamic BP	-1.3	6	3.7	8.5	Device size 64 mm × 46 mm Board size not

Graphene tattoo ⁴ Bio-impedance	Not Extra connecting wires required from the graphene tattoo to the sampling board.	Yes 5+ hours long. 5-10s time window. Dynamic BP	0.2	4.5	0.2	5.8	reported. 200 nm in thickness. Extra sampling board required.
Piezoelectric pressure sensor ★	Yes Soft sensor together with flexible sampling board encapsulated in a thin wristband.	Yes All day. Beat by beat. Dynamic BP	0.11	3.68	-0.05	4.61	Planar size: 15 cm × 35 cm Thickness 4 mm

★ This work.

Newly added references:

13. Byfield, R., Miller, M., Miles, J., Guidoboni, G. & Lin, J. Towards Robust Blood Pressure Estimation From Pulse Wave Velocity Measured by Photoplethysmography Sensors. *IEEE Sens. J.* 22, 2475–2483 (2022).
41. Yi, Z. et al. Piezoelectric Dynamics of Arterial Pulse for Wearable Continuous Blood Pressure Monitoring. *Adv. Mater.* 34, 2110291 (2022).
51. Seo, J., Pietrangelo, S. J., Lee, H.-S. & Sodini, C. G. Noninvasive arterial blood pressure waveform monitoring using two- element ultrasound system. *IEEE Trans. Ultrason. Ferroelectr. Freq. Control* 62, 776–784 (2015).
52. Huang, J.-J., Huang, Y.-M. & Chang, M.-W. Using bioimpedance plethysmography for measuring the pulse wave velocity of peripheral vascular. in 2016 13th International Conference on Electrical Engineering/Electronics, Computer, Telecommunications and Information Technology (ECTI-CON) 1–5 (2016). doi:10.1109/ECTICon.2016.7561449.
53. Nabeel, P. M., Jayaraj, J. & Mohanasankar, S. Single-source PPG-based local pulse wave velocity measurement: a potential cuffless blood pressure estimation technique. *Physiol. Meas.* 38, 2122–2140 (2017).

Comment 1: A study on converting the response into a BP waveform using bulky piezoelectric materials was published in *Adv. Mater.*, 34, 2110291, 2022. Compared to previous papers, there is no novelty in terms of material or BP waveform calculation process, and only the BP estimation process was changed, followed by post-processing.

Our response: We thank the referee for the valuable comment. We feel we may not clearly identify the difference between our work and this work in the first version of manuscript. There are several differences in the aspects of sensing accuracy, sensing adaptability, system integration level, device stability, and BP estimation performance. It is clear all of these points are significant for wearable continuous BP measurement.

From the points of device design and system integration, we presented a highly integrated system, with sensor array, back pressure adaption module, and signal

processing module integrated in a totally wearable format. In comparison, the reference (Adv. Mater., 34, 2110291, 2022) only presented a wearable sensor, fixed on skin with the assistance of tapes. The rigid signal sampling module was not integrated with sensors, and thus is not suitable for wearable measurement.

From the point of device stability and adaptability, the sensor in the reference was roughly fixed with a tape, where the interfacial stability needs to be addressed for long-term monitoring. In addition, encapsulation will further increase the interfacial instability, which is also a common issue in skin-integrated electronics. Based on this consideration, we presented a universal strategy, that is the active pressure adaption module to address the interfacial performance of skin-integrated electronics.

From the point of BP estimation performance, the BP calibration method in the reference relied on the calibration factor between the measured pressure wave and continuous BP wave, where repeated calibration process is required before each measurement. In comparison, our pretrained model provide a better BP estimation performance without repeated calibration process required.

Modifications: We added the reference (Adv. Mater., 34, 2110291, 2022) in **Supplementary table 1** and compared its performance with ours in terms of wearability, continuous, measurement accuracy and device dimension.

In Line 46, supplementary, we added comments to show the comparison between published works (wearable sensors/BP estimation models) and our work. The modified detail can be found in response of the reviewer's over all comments above.

Newly added references:

41. Yi, Z. et al. Piezoelectric Dynamics of Arterial Pulse for Wearable Continuous Blood Pressure Monitoring. Adv. Mater. 34, 2110291 (2022).

Comment 2: Papers such as "IEEE Sensors Journal, 21,11,12498-12510, 2021," and "IEEE Sensors Journal, 22, 3, 2475-2483, 2022," have measured BP by machine learning based on PWV values and multiple physiological features. There is no significant difference in the BP calculation method compared to previous papers. This system does not show any improvement like higher accuracy in BP estimation and sensor sensitivity.

Our response: We thank the referee for the comment. We summarized our contributions in terms of wearability, measurement accuracy, and model robustness that are quite different from the reference IEEE Sensors Journal, 21,11,12498-12510, 2021.

From the point of wearability and system integration, although traditional approach utilized multiple sensors (ECG and PPG) to calculate PWV and multiple features to estimate BP, multiple sensors located in two arteries, normally the femoral artery and

the carotid artery, greatly increase the system bulkiness. In comparison, our system provided a totally wearable interface with two sensor units 15 mm apart from each other to calculate local PWV for the BP estimation, which greatly increases its wearability.

From the point of measurement accuracy, traditionally, ECG sensor and PPG sensor were utilized to calculate the PWV, where the poor data synchronization between two sensors reduced the measurement accuracy. In addition, many of the results of the BP estimation in the above-mentioned references utilized the clean data of online database, which is recorded in a special ICU environment and required strictly recording position. In comparison, we sampled the measured pulse waves from two sensor units with a single ADC, which can effectively address the data synchronization issue. Besides, due to the presence of common noise, our method exhibits certain robustness and offer good accuracy compared to other continuous BP estimation methods.

Modifications: We added these related references in **Supplementary table 1** and compared their performance with ours in terms of in terms of wearability, continuous, measurement accuracy and device dimension. Besides, we summarized existing data models for BP estimation in supplementary note 1.

In Line 46, supplementary, we added comments to show the comparison between published works (wearable sensors/BP estimation models) and our work. The modified detail can be found in response of the reviewer's over all comments above.

Newly added references:

13. Byfield, R., Miller, M., Miles, J., Guidoboni, G. & Lin, J. Towards Robust Blood Pressure Estimation From Pulse Wave Velocity Measured by Photoplethysmography Sensors. *IEEE Sens. J.* 22, 2475–2483 (2022).

Comment 3: Does measuring the pulse wave velocity (PWV) at a short distance have medically significant results? Conventional PWV is measured between two arteries that are far apart. Therefore, PWV values at distant points are considered to better reflect cardiovascular parameters. Further explanation is necessary to show that the local PWV reflects cardiovascular parameters medically.

Our response: We thank the referee for this comment. Clinically, global PWV is commonly measured between femoral artery and carotid artery, which has been regarded as an important indicator of elastic property of the arterial system. However, global PWV only provides the average speed over a long segment composed of arteries with different mechanical properties^{47,48}. Consequently, global PWV might mask the initial variations in biomechanical properties in a small segment of arteries. In comparison, local PWV is measured on a short segment of a single artery, and thus can effectively reflect and identify the local stiffness of arterial wall⁴⁹. We added the explanation in manuscript to facilitate better understanding of global PWV and local

PWV, and why local PWV is adopted in this work.

In addition, it was reported that PWV is highly related with BP, and thus it has been adopted for indirect BP estimation. In this work, we adopted local PWV as an indicator to improve the estimation accuracy of our data model. To confirm this, we conducted experiments to compare the BP estimation accuracy with and without local PWV involved in model training. The estimation accuracy without PWV involved is presented in **Supplementary Fig. 13**, with an accuracy of 5.32 ± 11.42 mmHg and 2.78 ± 4.10 mmHg for SBP and DBP, respectively. We added the explanation in manuscript to illustrate the improved estimation accuracy with local PWV involved.

Modifications: In Line 244, manuscript, we added explanation on the significance of local PWV and modified the text as “In addition, blood propagation speed is also an important parameter that is highly related to the elasticity of the blood vessels, and thus can serve as a key determinant for BP^{27,44}. PWV has been widely used^{45,46} to improve the measurement accuracy of BP. However, global PWV reflects the elastic properties of arteries over a long segment, which might mask the initial variations in biomechanical properties in a small segment^{47,48}. Moreover, incorrect estimation of blood travel distance would cause accuracy issue. In comparison, local PWV calculating pulse wave velocity in a short distance along arteries provides important diagnostic information of biomechanical properties for local artery walls⁴⁹. However, the high propagation speed, normally ranging from 2 m/s to 20 m/s^{50–52}, as well as the short propagation distance between two adjacent sensing units bring an issue of short time difference (Δt) between two sensors’ responses, and therefore requiring a high sampling rate⁵³ (> 1500 data points/s) to reduce the error in Δt calculation, which arises a big challenge for wearable wireless transmission.”

In Line 323, manuscript, we added explanation on the performance comparison between data models with and without local PWV involved in the model training process and modified the text as “The testing results show the XGBoost model performed the highest accuracy among all the tested models, with an accuracy of -0.048 ± 4.613 mmHg for SBP and 0.108 ± 3.681 mmHg for DBP, which reached a Grade A according to the British Hypertension Society (BHS) standard⁶¹ (**Fig. 3d**, **Supplementary Fig. 12**), significant higher than that without PWV involved (**Supplementary Fig. 15**).”

Supplementary Fig. 15. SBP (a) and DBP (b) estimation accuracy without local PWV involved in model training.

Newly added references:

47. Vappou, J., Luo, J., Okajima, K., Di Tullio, M. & Konofagou, E. Aortic pulse wave velocity measured by pulse wave imaging (PWI): A comparison with applanation tonometry. *Artery Research* 5, 65–71 (2011).
48. Pereira, T., Correia, C. & Cardoso, J. Novel Methods for Pulse Wave Velocity Measurement. *J. Med. Biol. Eng.* 35, 555–565 (2015).
49. Darwich, M. A., Langevin, F. & Darwich, K. Local Pulse Wave Velocity Estimation in the Carotids Using Dynamic MR Sequences. *Journal of Biomedical Science and Engineering* 8, 227–236 (2015).

Comment 4: In the text, it was expressed that "grade A according to the British Hypertension Society." The criteria for Grade A are shown in the table only as average and standard deviation values. Additionally, there is no standard for the total number of subjects to be measured. The sample size of 17 subjects is insufficient to demonstrate the reliability of the measurement. International standards require a minimum of 85 subjects using an auscultatory sphygmomanometer as a reference.

Our response: We thank the referee for this comment. In revision, we further tested another 70 volunteers, a total number of 87 volunteers including the previous 17 volunteers, and analyzed the measurement accuracy according to the BHS protocol to evaluate the measurement accuracy of our devices. **Supplementary Fig. 20** shows the measurement accuracy comparison with a commercial sphygmomanometer (Omron HEM-7156T) as a reference, where continuous BP data for 1 minute was measured with our device to calculate the measured mean BP and standard deviation. From **Supplementary Fig. 20**, it is obvious that the error of all the 87 volunteers is less than 15 mmHg regardless of SBP or DBP. The error less than 10 mmHg and 5 mmHg take 98.9% and 88.5% for DBP, and take 95.4% and 80.5% for SBP, which achieves Grade

A accuracy level based on BHS protocol for evaluation of BP monitoring devices. Moreover, we performed continuous BP evaluation with continuous measured by commercial continuous BP monitoring system (CNAP) as a reference to evaluate the measurement accuracy of our device. **Extended Data Figure 9** presents the error distribution of all the 87 volunteers and statistical results of the measurement error across different BP categories recommended by the American Heart Association, from which 98% accuracy can be achieved in less than 15 mmHg level and over 70% accuracy can be achieved in less than 5 mmHg level. In summary, all these results on the 87 volunteers indicate the great accuracy of our device.

Modifications: In Line 363, manuscript, we added description of on statistical evaluation of 87 volunteers and modified the text as “To validate the measure accuracy of the TSMS system, a total number of 87 volunteers were selected to continuously monitor their BP for 2 minutes with BP measured by commercial noninvasive arterial blood pressure monitor (CNAP) and commercial sphygmomanometer (Omron HEM-7156T) as references. **Extended Data Fig. 9a** shows the statistic error distribution of the 87 volunteers, where most of error bars are less than 10 mmHg regardless of SBP or DBP. Moreover, quantitative statistics on SBP and DBP show that excellent measurement accuracy was achieved with the TSMS, with over 70% for error < 5 mmHg and over 98% accuracy for error < 15 mmHg (**Extended Data Fig. 9b, 9c, Supplementary Fig. 20**).”

Supplementary Fig. 20. Measurement accuracy comparison in DBP (a) and SBP (b) with commercial sphygmomanometer as a reference.

Extended Data Figure 9. Performance validation of the TSMS on a total number of 87 volunteers with commercial continuous BP monitoring system as a reference. (a). Violin plot representing the 2 minutes continuous systolic BP and diastolic BP distribution of the 87 volunteers. Measurement accuracy statistics on systolic BP (b) and diastolic BP (c) representing the measurement accuracy in different BP categories with continuous BP measured by CNAP as reference.

Comment 5: A micro-airbag is included with the sensor to overcome the conformal attachment to the skin due to mechanical stiffness from bulky piezoelectric materials. However, there may be distortion in the measured waveform due to the pressure applied by the micro-airbag. In the text, it was expressed that the pressure of 0-12 kPa (0-90 mmHg) with a micro-airbag is less than the systolic BP value, making it suitable for continuous monitoring without discomfort. However, even if the systolic BP is over 100 mmHg, the pressure felt on the skin is much smaller than this, so the effect of the pressure from the micro-airbag is greater than the actual arterial movement on the skin.

Therefore, data showing no distortion of the BP waveform for various micro-airbag pressures should be provided."

Our response: We thank the referee for these insightful suggestions. For the first question, we conducted experiments to study the influence of back pressure levels on the pulse waveform and measurement accuracy. **Extended Data Figure 8c and 8d** show the comparison on measured pulse wave and statistical BP under different backpressure, ranging from 0 to 5 pumping phases. From **Extended Data Figure 8c**, although increased signal amplitude was measured with increased backpressure level, there is no obvious changes in SBP and DBP observed in **Extended Data Figure 8c**, with changes in average less than 2 mmHg. As a result, we can conclude that the changes in backpressure did not reduce BP measurement accuracy.

For the second question, individual differences, such as BMI and bone structure, determine the strength of the pulse wave, and further the minimum backpressure level for effective measurement of pulse wave. To study if high backpressure level will generate discomfort feelings to users, we conducted new experiments to study the minimum pressure for effective detecting of pulse wave signal for volunteers with different BMI and recorded if discomfort generated for 30 minutes continuous measurement. From **Supplementary Fig. 7**, the minimum pressure required for most of volunteers less than 2 pumping phases and only one volunteers with BMI over 24 required 3 pumping phases for effective detecting of pulse wave. Besides, there is no discomfort feeling generated for all the six volunteers after 30 minutes continuous measurement, from which we can conclude that the active pressure adaption module will not cause discomfort to users.

Modifications: In Line 338, manuscript, for the influence of applied back pressure on the measurement accuracy, we added explanation on the experiments results and modified the text as "To study the improvement of the active pressure adaption module on measurement accuracy and system robustness, the influence of pressure changes in the micro airbag on measurement accuracy was studied. **Extended Data Fig. 8a and 8b** present the continuous BP wave comparison between the micro airbag just pumped and 30 minutes after the inflation, where SBP and DBP show small fluctuations between 0 minute and 30 minutes of the measurement. Besides, the micro airbag was inflated with different pressure, ranging from 0 to 5 pumping phases, to study the influence of the changes in backpressure on measurement accuracy, as shown in **Extended Data Fig. 8c and 8d**, where SBP and DBP show small fluctuations with average BP fluctuates no more than 2 mmHg."

In Line 185, manuscript, we added explanation on the minimum backpressure level and if discomfort generated during tests and modified the text as "The pressure adaptation module serves a powerful support to the sensor for increasing the sensing signal quality. However, too high back pressure may also cause discomfort to users due to the closure of artery. Therefore, we set the maximum pressure at 12 kPa (90 mmHg) to avoid

discomfortable feelings while maintaining satisfactory signal quality (Supplementary Fig. 7).”

Extended Data Figure 8. BP estimation performance validation and stability improvement with the active pressure adaption module. Continuous BP wave forms (a) and statistical BP (b) comparison for 1 minute between the micro airbag just pumped and 30 minutes after the inflation. Comparison on measured continuous pulse waves (c) and statistical BP (d) under different airbag pressure. Signal stability comparison between with (e) and without backpressure (f) applied.

Supplementary Fig. 7. Minimum pumping phases required for effective detecting of continuous pulse wave and if discomfort feeling generated during 30 minutes continuous measurement.

Comment 6: Further revisions are required to enhance the linguistic and grammatical precision of the entire manuscript, with particular emphasis on achieving consistent adherence to the conventions pertaining to units and spacing.

Our response: We thank the referee for the valuable comment. We have carefully addressed the linguistic and grammatical issues throughout the manuscript and uniformed the spacing between units and numbers.

Modifications: We have carefully addressed the linguistic and grammatical issues throughout the manuscript and uniformed the spacing between units and numbers.

Responses to comments of Referee #2

Comments from Referee #2:

Summary Comment: This work reported a continuous blood pressure monitoring strategy with the ability of active pressure adaption and in-situ transformation of sensory signals. Such a thin, soft, miniaturized system can achieve a high measurement accuracy of 0.11 ± 3.68 mmHg DBP and -0.05 ± 4.61 mmHg SBP, which has reached the Grade A level of standard wearable BP monitoring devices. This work will definitely be a remarkable promotion into this field and will attract broad attention in the field of CVDs. To improve this manuscript, some personal suggestions are provided for further discussion and some minor concerns should be addressed before the publication of this paper.

Our response: We thank the referee for these positive comments. We carefully addressed these issues and revised the manuscript accordingly.

Modifications: None

Comment 1: L52. In the introduction, to my knowledge, the mean blood pressure may be a calculated value based on the formula $MAP = [SBP + (2 \times DBP)]/3$ and not a recording index in clinic.

Our response: We thank the referee for the useful suggestion. The mean BP is not a recording index in clinic. Commonly, only SBP and DBP are measured clinically, and we can calculate the MAP based on the formula $MAP = [SBP + (2 \times DBP)]/3$. We revised the expression in manuscript.

Modifications: In Line 53, we modified the expression as “At present, clinics mainly rely on the cuff-based sphygmomanometers to measure the static BP, in which only isolated casual systolic BP (SBP), diastolic BP (DBP) are recorded, and mean BP (MAP) can be calculated according to SBP and DBP.”

Comment 2: Grade A should be provided with one reference or a detailed description for readers’ better understanding. Which standard in the British Hypertension Society (BHS) is referred to for this evaluation?

Our response: We thank the referee for these comprehensive suggestions. We added a “BHS protocol for the evaluation of blood pressure measuring devices.” as a reference for better understanding of BP accuracy level.

Modifications: In Line 323, manuscript, we modified the text as “The testing results show the XGBoost model performed the highest accuracy among all the tested models,

with an accuracy of 0.002 ± 3.335 mmHg for SBP and 0.025 ± 4.582 mmHg for DBP, which reached a Grade A according to the British Hypertension Society (BHS) standard⁶³ (**Fig. 3d, Supplementary Fig. 14**), significantly higher than that without PWV involved (**Supplementary Fig. 15**).

Newly added reference:

63. O’Brien, E., Atkins, N., Mee, F. & O’malley, K. Evaluation of Blood Pressure Measuring Devices. Clin. Exp. Hypertens. 15, 1087–1097 (1993).

Comment 3: For a better reading, supplementary table 1 is suggested to be further clarified. For example, some sensors are described in terms of the method. A consistent unit should be better for understanding the mentioned devices' dimensional differences. How to define dynamic and static BP and why is the continuous feature classified into static and dynamic BP? It is better to provide the type of commercial products if it is summarized in the table.

Our response: We thank the referee for these useful suggestions. We further clarified the references in Supplementary table 1 based on their devices instead of the measurement methods, and we unified the units in the description of device dimension.

For the definition of dynamic and static BP, it refers to a concept of BP variability that reveals the dynamic BP fluctuations resulted from environmental, physical and emotional factors. Therefore, dynamic BP fluctuations have been created by changing the environmental factors to evaluate if the devices can accurately track these dynamic BP changes. We added the definition of the dynamic and static BP in the supplementary table 1.

For the commercial products, to the best of our knowledge, there is no wearable products for continuous BP measurement in the market. Therefore, there is no commercial products listed in Supplementary table 1.

Modifications: In Line 395, supplementary, we added the description of dynamic BP and static BP. “Dynamic BP and static BP are defined as whether BP fluctuations were created during the measurement.” Besides, we further clarified the references based on their devices instead of their measurement methods and unified the units of devices dimension column.

Supplementary Table 1. Technical comparison of continuous BP monitoring.

Method	Wireless? Totally wearable?	Continuous?	Accuracy				Device dimension
			DBP (mmHg)		SBP (mmHg)		
			ME	SD	ME	SD	
PPG ¹¹	Yes	Not	-0.07 ± 4.47		0.00 ± 3.61		Not

	PPG sensor and hard sampling board in a watch shaped wrist.							reported.
PPG ⁸	Not Commercial PPG sensor and ECG electrode required.	Yes Beat to beat. Static BP.	3.23 ± 4.75		4.43 ± 6.09			Not reported.
PPG ¹²	Yes	Yes 1000s long. Continuous. Static BP.	Not reported.		0.24	1.18		Device size 4 cm × 2 cm Board size Not reported.
PPG ¹³	Yes Neither the sensor nor the electrical circuit are designed in wearable format.	Not	2.12	0.26	2.94	0.72		Not reported.
Piezoelectric ultrasound transducer ¹⁴	Not Connecting to sampling equipment required.	Yes 30s long. Continuous. Static BP.			Not reported.			23 mm × 20 mm
Resistive pressure sensor combined with ECG ¹⁰	Yes Wearable pulse sensor and ECG electrodes are worn separately.	Yes 4 hour long. 15s time window. Static BP.	0.24	5.19	0.07	9.66		ECG patch 7 cm × 2 cm Pulse sensor 4 cm × 1.7 cm
Piezoelectric pressure sensor ³	Not Sensor in wearable format. Rigid sampling board. Sensor and electrical board were not integrated.	Yes Static BP.	Not reported.		Not reported.			Device size 3.5 mm in diameter. Board size not reported.
Capacitive pressure sensor ¹⁵	Not Connecting to sampling equipment required.	Yes 10 min. long. Static BP	0.48 ± 1.96		1.43 ± 1.96			8 mm × 8 mm
Capacitive pressure sensor ¹⁶	Not Fixing wristband and sampling equipment required.	Yes 9 min. long. 70 beats window. Static BP			-0.054 ± 2.09			15 mm × 2 mm
Commercial silver electrodes. Bio-impedance ⁹	Not Bulky wristband and sampling board required.	Yes 60 min. long. 5-10s time window. Dynamic BP	-1.3	6	3.7	8.5		Device size 64 mm × 46 mm Board size not reported.
Graphene tattoo ⁴ Bio-impedance	Not Extra connecting wires required from the graphene tattoo to the sampling board.	Yes 5+ hours long. 5-10s time window. Dynamic BP	0.2	4.5	0.2	5.8		200 nm in thickness. Extra sampling board required.
Piezoelectric pressure sensor ★	Yes Soft sensor together with	Yes All day. Beat by beat.	0.11	3.68	-0.05	4.61		Planar size: 15 cm × 35 cm

★ This work.

Dynamic BP and static BP are defined as whether BP fluctuations were created during the measurement.

Comment 4: Regarding the airbag system, obtaining stable pressure in the airbags is critical for achieving high accuracy in BP monitoring in my opinion. Generally, maintaining such a stable pressure requires a closed-loop system, including an air pump, a pressure sensor, and valves. In this work, the pressure in the airbag is fluctuant as I understand. The authors should address why this design is preferred to some traditional closed-loop systems. The other question is about the motion artifact, which is the most critical challenge in continuous BP monitoring at present. How to consider it based on this design.

Our response: We appreciate the referee for these insightful comments. Obviously, the referee has rich experimental experiences in BP measurement and control system. From the point of stabilizing the pressure in the airbag, traditional close-looped system, with an extra pressure sensor applied, would be a better choice to control the pressure in the airbag. In practice, close-looped pressure control would not only increase the system complexity due to the presence of pressure sensor but greatly increase the power consumption as frequently pumping phases required. In this work, we aimed to enhance the interfacial performance of the pressure sensor to improve signals quality for BP estimation. Therefore, we developed a peak detection algorithm to build close-looped feedback between the micro-pump and measured pulse wave. Obviously, the pressure fluctuations in the airbag might arise BP estimation error. To find a balanced control strategy, we conducted experiments to compare the influence of pressure fluctuation on BP estimation accuracy. **Extended Data Figure 8a and 8b** and **Supplementary Fig. 18** show the comparison in measured pulse wave and estimated BP between the micro airbag just pumped and 30 minutes after the inflation. From **Supplementary Fig. 18**, we can see decreased signal amplitude measured due to decreased pressure level in the micro airbag. From **Extended Data Figure 8a and 8b**, we can see that the pressure fluctuations in the micro airbag did not change the estimated BP, with small fluctuations in average SBP and DBP. Moreover, we conducted experiments to study the influence of back pressure levels on the pulse waveform and measurement accuracy. **Extended Data Figure 8c and 8d** show the comparison on measured pulse wave and statistical BP under different backpressure, ranging from 0 to 5 pumping phases. From **Extended Data Figure 8c**, although increased signal amplitude was measured with increased backpressure level, there is no obvious changes in SBP and DBP observed in **Extended Data Figure 8c**, with changes in average less than 2 mmHg. In summary, from these experimental results, we can conclude that the fluctuations in backpressure levels did not reduce the measurement accuracy. We added explanation in the manuscript to facilitate better understanding.

For the motion artifact, to investigate how the active backpressure adaption module improve the interfacial performance against motion artifact, we asked user to wear our device on the wrist and waved his forearm. The measured pulse wave is shown in **Extended Data Figure 8c** and **8d**, from which signal distortion was generated when without backpressure applied in **Extended Data Figure 8c**. In **Extended Data Figure 8d**, although fluctuations in signal amplitude generated by forearm movement, the measured pulse wave maintained. From the result, we can conclude that the active pressure adaption module can effectively improve interfacial performance against motion artifact.

Modifications: In Line 338, manuscript, we added explanation on the influence of pressure fluctuation on measurement accuracy and how the backpressure improve the interfacial performance against motion artifact. We modified the text as “To study the improvement of the active pressure adaption module on measurement accuracy and system robustness, the influence of pressure changes in the micro airbag on measurement accuracy was studied. **Extended Data Fig. 8a** and **8b** present the continuous BP wave comparison between the micro airbag just pumped and 30 minutes after the inflation, where SBP and DBP show small fluctuations between 0 minute and 30 minutes of the measurement. Besides, the micro airbag was inflated with different pressure, ranging from 0 to 5 pumping phases, to study the influence of the changes in backpressure on measurement accuracy, as shown in **Extended Data Fig. 8c** and **8d**, where SBP and DBP show small fluctuations with average BP fluctuates no more than 2 mmHg. **Extended Data Fig. 8e** and **8f** show the comparison in measured pulse wave with and without backpressure applied under forearm waving, where obvious signal distortion was generated when backpressure was not applied. In comparison, there is no signal distortion generated when backpressure was applied, with only fluctuations in signal amplitude generated response to forearm movement. It is obvious that the introduction of the active pressure adaption module significantly improves the system robustness against body movement while maintaining measurement accuracy compared to those without backpressure applied.”

Supplementary Fig. 18. Comparison in measured pulse wave at 0 and 30 minutes after the inflation.

Extended Data Figure 8. BP estimation performance validation and stability improvement with the active pressure adaption module. Continuous BP wave forms (a) and statistical BP (b) comparison for 1 minute between the micro airbag just pumped and 30 minutes after the inflation. Comparison on measured continuous pulse waves (c) and statistical BP (d) under different airbag pressure. Signal stability comparison between with (e) and without backpressure (f) applied.

Comment 5: Extended Data Figure 5f, force(mV)? How about the sensitivity of the used piezoelectric pressure sensor?

Our response: We thank the referee for these useful comments. For the unite of force in **Extended Data Figure 5f**, we modified it to (mN). For the sensitivity of the

piezoelectric pressure sensor, we calculated the sensitivity of the sensor based on the output in **Extended Data Figure 4a**, corresponding to 0.2 V/N with loading frequency at 1 Hz. We added the sensitivity data in the the manuscript.

Modifications: In Line 827, manuscript, we added the sensitivity in the legend of Extended Fig. 4 and modified the legend as “**Extended Data Figure 4. Electric characterization of the piezoelectric sensor.** (a). Output voltage versus loading force of the piezoelectric sensor in the force range of 5-35mN. The sensor presents excellent linearity with force sensitivity at 0.2 V/N. (b). Corresponding output voltage with the loading force of 32 mN at 1Hz. (c). Output current of versus loading force of the piezoelectric sensor in series with an external resistance, with resistance value at 360 K Ω . (d). Corresponding current with the loading force of 32 mN at 1Hz. (e). Output voltage of the piezoelectric sensor under the loading force of 50 mN over 2500 loading cycles. Inset figures show the excellent stability of the sensor, with stable output voltage generated after over 2500 loading cycles.”

Extended Data Figure 4. Electric characterization of the piezoelectric sensor. (a). Output voltage versus loading force of the piezoelectric sensor in the force range of 5-35 mN. The sensor presents excellent linearity with force sensitivity at 0.2 V/N. (b).

Corresponding output voltage with the loading force of 32 mN at 1Hz. (c). Output current of versus loading force of the piezoelectric sensor in series with an external resistance, with resistance value at 360 K Ω . (d). Corresponding current with the loading force of 32 mN at 1 Hz. (e). Output voltage of the piezoelectric sensor under the loading force of 50 mN over 2500 loading cycles. Inset figures show the excellent stability of the sensor, with stable output voltage generated after over 2500 loading cycles.

Extended Data Figure 5. Theoretical and experiments characterization of the sensor output on soft and rigid substrate. (a). Schematic illustration of the Micro Newton Tester and device deformation on soft substrate and rigid substrate. (b) and (c). Optical images presenting the device with glass substrate (b) and PDMS substrate (c) loaded by the tester. (d). FEA result presenting the electric potential on the top surface of the piezoelectric transducer under the loading force of 100mN. (e). FEA result showing the spatial displacement of the sensor and substrate under the loading force of 100mN. (f) and (g). Results comparison between theoretical simulation and experiments of the surface electric potential (f) and spatial displacement (g) under a range of loading force ranging from 0 to 100mN, where the voltage outputs show excellent consistency under the loading of pulse propagation, ranging from 0 to 20 mN.

Comment 6: About the “denoise the signal with a cutoff frequency of 0.5 to 8Hz”, is it possible to provide a detailed discussion of why the cutoff frequency is 0.5 to 8 Hz?

Our response: We thank the referee for the valuable comment regarding the cutoff frequencies used to denoise the signal. Our approach of using a cutoff frequency of 0.5 to 8 Hz was chosen after consideration of remaining the main waveform features of the measured pulse signal and reduce the impact of noise on model accuracy. To remove baseline drift, we removed low frequency signals below 0.5 Hz, while filtering out high frequency signals above 8 Hz, such as powerline interference and some motion artifacts. Doing so allowed us to retain the main components of the signal while minimizing noise. Such a filter setting was also adopted by several published works (Sensors 2021, 21, 7233; Sensors 2019, 19, 3420).

Modifications: In Line 270, manuscript, we add two references and explain the filter setting and modified the text as “It is possible to get BP values from the raw pulse waveform, but respiration and body movements could lead to distortion of the signal waveform, thus affecting the accuracy^{55,56}. Therefore, we used a third-order IIR bandpass filter to denoise the signal with a cutoff frequency of 0.5 to 8 Hz, which has been widely adopted to remove baseline drift and high frequency noise^{57,58}.”

Newly added references:

57. Slapničar, G., Mlakar, N. & Luštrek, M. Blood Pressure Estimation from Photoplethysmogram Using a Spectro-Temporal Deep Neural Network. *Sensors* 19, 3420 (2019).
58. Ramesh, J., Solatidehkordi, Z., Aburukba, R. & Sagahyroon, A. Atrial Fibrillation Classification with Smart Wearables Using Short-Term Heart Rate Variability and Deep Convolutional Neural Networks. *Sensors* 21, 7233 (2021)

Comment 7: The output voltage of the piezoelectric sensor seems to be not stable as the authors described. The authors should provide a detailed discussion or necessary evidence about what will be caused by this fluctuation in BP monitoring and whether it can be mediated by the loads applied by the airbags.

Our response: We thank the referee for these insightful comments. The measured piezo response might present instability, commonly composed of noise and baseline drift. To investigate the robustness of the piezo sensor, and if the stability was improved with our active pressure adaption module, we conducted experiments to compare the measured piezo response with and without active pressure applied under two typical situations. **Extended Data Figure 8c** and **8d** present the comparison of measured pulse wave with and without backpressure applied. From **Extended Data Figure 8c**, signal fluctuations can be found without body movement, and signal distortion was generated under forearm waving (highlighted with red dotted rectangular). When backpressure

was applied (**Extended Data Figure 8d**), signal quality was greatly improved, without distortion generated with body movement only fluctuations in signal amplitude.

Modifications: In Line 338, manuscript, we added explanation on the improvement of signal quality with backpressure applied and modified the text as “To study the improvement of the active pressure adaption module on measurement accuracy and system robustness, the influence of pressure changes in the micro airbag on measurement accuracy was studied. **Extended Data Fig. 8a** and **8b** and **Supplementary Fig. 18** present the continuous BP wave comparison between the micro airbag just pumped and 30 minutes after the inflation, where SBP and DBP show small fluctuations between 0 minute and 30 minutes of the measurement. Besides, the micro airbag was inflated with different pressure, ranging from 0 to 5 pumping phases, to study the influence of the changes in backpressure on measurement accuracy, as shown in **Extended Data Fig. 8c** and **8d**, where SBP and DBP show small fluctuations with average BP fluctuates no more than 2 mmHg. **Extended Data Fig. 8e** and **8f** show the comparison in measured pulse wave with and without backpressure applied under forearm waving, where obvious signal distortion was generated when backpressure was not applied. In comparison, there is no signal distortion generated when backpressure was applied, with only fluctuations in signal amplitude generated response to forearm movement. It is obvious that the introduction of the active pressure adaption module significantly improves the system robustness against body movement while maintaining measurement accuracy compared to those without backpressure applied.”

Extended Data Figure 8. BP estimation performance validation and stability improvement with the active pressure adaption module. Continuous BP wave forms (a) and statistical BP (b) comparison for 1 minute between the micro airbag just pumped and 30 minutes after the inflation. Comparison on measured continuous pulse waves (c) and statistical BP (d) under different airbag pressure. Signal stability comparison between with (e) and without backpressure (f) applied.

Comment 8: In the main section, the description of the signal processing module is vague, could you explain its functions?

Our response: We thank the referee for these useful comments. The data processing module mentioned is the data processing in MCU before transmitting to mobile device.

The signal collected from two sensing units; we extracted the time interval of piezo response and down sample the signal before transmission to reduce the amount of data transmission.

Modifications: In Line 87, manuscript, we detailed the description on signal processing module and modified the text as “The TSMS composes three subsystems, including a sensing module to detect the blood pulse wave, an active pressure adaptation module to provide powerful back pressure for improved interfacial properties, and a data processing module to in-situ extract pulse transit time interval and transmit the measured data to a graphical user interface (GUI).”

Comment 9: L117-121. The sentence is hard to read. I suggest rephrasing it “The pair of PZT sensors is encapsulated with soft silicone, providing soft and conformal contact between the sensors and skin, thus allowing for accurate conversion of local deformation of the sensor caused by the expansion/contraction of the artery into electrical outputs.”

L125-127. I prefer the below description: “By measuring the pulse travel time (PTT) of a single pulse waveform arriving at two sensors, and the distance between the two electrodes, the local Pulse Wave Velocity (PWV) can be calculated.”

Our response: We thank the referee for these useful comments. After carefully reviewing the manuscript, we revised all the listed issues.

Modifications: In Line 118, we modified the text as “The pair of the PZT sensors is encapsulated with soft silicone (Polydimethylsiloxane, PDMS, 145 kPa), providing soft and conformal contact between the sensors and the artery located skin region, thus allowing for accurate conversion of local deformation of the sensor caused by the expansion/contraction of the artery into electrical outputs.”

In Line 127, we modified the text as “Meanwhile, by measuring the pulse travel time (PTT) of a single pulse waveform arriving at two sensors, and the distance between them, the local PWV can be calculated (**Fig. 1a, middle**).”

Comment 10: L146. Please be more specific. If the measurement of “6 times measures per hour” is intermittent, and how long each measurement takes?

Our response: We thank the referee for the important suggestion. We added explanation on the measurement duration in the text.

Modifications: In Line 147, we modified the text as “The wireless system could continuously work for almost two days (6 times measures per hour, and each measurement lasts 15 s to get continuous BP, **Supplementary Fig. 5**).”

Comment 11: In Supplementary Fig. 5, the figure legend of Supplementary Fig. 5(d)

doesn't match the data presented in the figure, please revise it.

Our response: We thank the referee for the useful comment. We modified the figure legend of Supplementary Fig. 5.

Modifications: In Line 247, supplementary, we modified the legend of Supplementary Fig. 5 as “**Supplementary Fig. 5. Lifetime of the rechargeable lithium-ion battery in intermittent working mode.** (a). Voltage variation data showing the working duration of almost two days. (b) and (c). Voltage variation data presenting working phases at the beginning and the ending of the test, respectively. (d). Voltage data showing the system is powered down after 33.5 h sampling.”

Supplementary Fig. 5. Lifetime of the rechargeable lithium-ion battery in intermittent working mode. (a). Voltage variation data showing the working duration of almost two days. (b) and (c). Voltage variation data presenting working phases at the beginning and the ending of the test, respectively. (d). Voltage data showing the system is powered down after 33.5 h sampling.

Comment 12: Figure2 h. The figure shows the pulse wave of two sensors displayed a different amplitude. Could you please explain the reason?

Our response: We thank the referee for the insightful comment. Theoretically, the blood flow propagates along the arterial tree, with the propagation speed increases with the distance from heart. Meanwhile, the propagation amplitude weakens with increased propagation distance due to the present of damping and absorption of biological tissues. In **Fig. 2h**, we schematically presented the difference in amplitude of the proximal pulse wave and the distal one.

Modifications: In Line 260, manuscript, we added the explanation on the decreased amplitude in distal pulse wave and modified the text as “**Fig. 2h** schematically presents the proximal and distal pulse wave, where a decreased amplitude in distal wave presents due to damping and biological absorption along the artery tree.”

Comment 13: Could you please annotate the subscripts of the variables in the equations of the chapter “piezo response conversion” to facilitate understanding?

Our response: We thank the referee for the important suggestion. We reorganized the equations in **Supplementary Note 2** and added the explanation for subscripts to facilitate better understanding.

Modifications: In Line 71, supplementary, we modified the subscripts of the constitutive equations of piezoelectric materials and added explanation on the subscripts as “The electromechanical coupling behavior of the piezoelectric transducer is governed the following constitutive equations.

$$\sigma_{ij} = c_{kl}\epsilon_{ij} - (e_{ik})^T E_j \quad (1)$$

$$D_i = e_{ik}\epsilon_k + k_{ij}E_j \quad (2)$$

Where σ , ϵ , E and D stand for the stress, strain, electrical field and electrical displacement, respectively. c , e and k stand for the elastic array, piezoelectric coupling array and dielectric array, respectively. i, j, k , and l represent the spatial direction of the parameter matrix, with i and j taking the values of 1, 2, and 3, and k and l taking the values of 1, 2, 3, 4, 5, 6. For the typical piezoelectric material PZT, the constitutive equations can be expanded into following equations.”

Comment 14: According to the description of L249: The signal was resampled to 200 Hz, which does not match the frequency for preprocessing, please explain the reason.

Our response: We thank the referee for the important suggestion. According to Nyquist sampling theorem, applying the higher sampling rate can better restore the original signal. Therefore, we resample the signal to 1000 Hz before the filtering to get better quality signal.

Modifications: In Line 273, manuscript, we added explanation on the signal resample process and modified the text as “Before denoising, we resampled the signal to 1000 Hz for better signal restoration. Then we performed the integration calculation with the denoised signal (**Supplementary Fig. 13**). It was segmented into beat-to-beat waveforms for feature extraction with normalized amplitude (Window size of 1500 points with a 30% overlap).”

Comment 15: Please provide more information about how the system detects if a signal is available for control of the micropump.

Our response: We thank the referee for the valuable comment. To detect if a signal is available, we applied a signal quality assessment based on three main factors, the peak interval, its standard deviation, and signal kurtosis. The peak intervals were calculated from distance between each pulse peaks' index extracted using the peak detection algorithm (**Supplementary Note 4**). For signal quality assessment, we set the thresholds of these indices with empirical values and then compared the results of the current signal segment with thresholds. If the values beyond the thresholds, the system considers the current signal as unavailable, and the MCU sets the command to the air pump to inflate, as illustrated in **Extended Data Figure 6**. We added the explanation on the signal detection strategy in the manuscript.

Modifications: In line 169, supplementary, we added explanation on signal quality assessment and modified the text as

“Supplementary Note 4: Peak detection algorithm and signal quality assessment

After we acquire the noise-reduced signal that lasts for 5 seconds, we down sample the signal to 1000 points per second. First, we calculate the first-order differential of the sequence and assign the positive values, zero-equal and negative values of the result to -1, 0, and 1, respectively. Second, we calculate the first-order differential of the previous result and then record the indexes of the negative values. We compute the mean value ‘avg’ and standard deviation ‘sd’ of all points. To search for peaks from the recorded array, we set the time distance threshold to 700 between adjacent peaks based on the empirical value. Afterward, we traverse the peak indexes and retain the candidates $y(i)$ that satisfy the condition $y(i) - \text{avg} > 1.5 \times \text{sd}$; The subsequent processing consists of filtering out the peaks that are too close to each other (less than the distance), and the remaining points are the desired wave peaks. For signal quality assessment, we set the peak interval threshold T_p , their standard deviation threshold SD_p and the signal kurtosis threshold T_k according to empirical values, and compare the actual values with the thresholds to determine whether the signal is available. We consider the signal as available if the following criteria are met: the relative difference between the mean peak interval and T_p is less than 25%, and both the standard deviation and the kurtosis of the signal are below their respective thresholds.”

Responses to comments of Referee #3

Comments from Referee #3:

Summary Comment: Overall, the article is well-written and informative, providing insight into the importance of monitoring blood pressure and the development of wearable devices for continuous non-invasive blood pressure monitoring. The integration of materials, devices, mechanics, data processing methods, and integration strategies for the system is informative, and the advantages of the system are clearly outlined. However, it is recommended that certain issues related to professional and grammatical be addressed prior to publication.

Our response: We thank the referee for these positive comments. We carefully addressed these issues and revised the manuscript accordingly.

Modifications: None

Professional aspects

Comment 1: Line 91: Provide more detail on the methodology, it would be helpful to provide more detail on the methodology used to design and test the system. For example, what specific PZT materials were used, and how were they chosen?

Our response: We thank the referee for the valuable comments. We added explanations on the system design and material selection criterion in manuscript to facilitate better understanding of how the system was design and how the material was selected.

Modifications: In Line 92, manuscript, we modified the text as “Highly chemical/physical stability of the piezoelectric material PZT 5H enables the sensor array with excellent uniformity and robustness while maintaining superior sensitivity due to its high piezoelectric coupling coefficient⁴⁰.”

In Line 109, manuscript, we modified the explanation on structural design as “The pulse sensing system associates with a pair of sensors spacing 15 mm from each other, that adopts piezoelectric thin layers, lead zirconate titanate (PZT 5H) as the sensors (3 mm × 3 mm, 200 μm in thickness), as shown in **Fig. 1b**. The piezoelectric sensors are based on sandwiched top/bottom electrodes, where the top electrode is routed to the bottom substrate via a vertical connecting point (VCP) for optimized mechanical robustness (**Supplementary Fig. 3**) and ease of electrical connection, that endow the sensors adaption to various mechanical deformations (**Fig. 1e, Supplementary Fig. 3**). A bilayer of polyimide (PI, 25 μm thick) and gold (Au, 200 nm thick) that is designed in serpentine patterns to serve as the flexible electrodes and interconnects for the sensor array while realizing flexibility. (See methods for the details, **Fig. 1b**).” to facilitate better understanding.

Newly added references:

40. Yang, J. An Introduction to the Theory of Piezoelectricity. vol. 9 (Springer International Publishing, 2018).

Comment 2: Line 102: The main section ends with the potential benefits of continuous BP monitoring. However, it could benefit from a more explicit statement about the purpose or objective of the study, to give readers a better sense of what the article will be focused on.

Our response: We thank the referee for the valuable suggestion. We modified the structure of the main section to end with the issues we aim to address in this work, and further the potential promotion we brought in the field of continuous BP measurement.

Modifications: In Line 100, manuscript, we modified the text as “Compared to other cuffless BP monitoring devices (**Supplementary Table 1**), this work addresses the issues in system integration level, interfacial performance, and BP estimation model, thus showing great advantages in terms of wearability, continuance, dynamics, and most importantly, measurement accuracy (**Grade A** level), which will promote its application in continuous BP monitoring.”

Comment 3: Line 104: The article would benefit from a more structured approach. The authors could use headings and subheadings to organize the information and make it easier to follow in Design and working principle of the wireless integrated BP monitoring system section.

Our response: We thank the referee for the useful suggestion. We added two headings in the part of Design and working principle to better organize its structure to facilitate better understanding.

Modifications: In Line 105 and Line 122, manuscript, we added headings to reorganize the structure of Design and working principle of the TSMS part. We modified the text as “

Design of the device

Fig. 1a shows the schematic illustration of the ...

Working principles and system integration

Piezo response is generated due to the...

...
”

Comment 4: Line 675: Figure 1: The information provided in (g) about PPG and Ultrasound not sufficiently clear, please revise it.

Our response: We thank the referee for the useful comment. We revised Fig. 1g as follows.

Modifications: We removed the background of Fig. 1g.

Figure 1. Working principle and layouts of the wearable wireless continuous blood pressure monitoring system. (a). Schematic diagram of signal conversion from piezo response to continuous blood pressure that is presented in a mobile GUI. Physical distance between two sampling sites and time difference in two sensing units were utilized to calculate localized PWV. Pulse wave features, together with localized PWV were transmitted to data for the estimation of beat-to-beat BP. (b). Explosive view of the wireless wristband, with three subsystems, sensing module, force generation module and signal processing module. (c). Optical image of the wireless wristband worn on user's wrist joint. (d). Optical image of all the system components before sealed in the silicone wristband. (e) and (f). Optical images of the sensor array and micro airbag array suffering from mechanical deformations. (g). Technical comparison between our device and published works for continuous BP monitoring in terms of wearability, accuracy, dynamics, continuance and wireless.

Comment 5: Line 106, 150: Use consistent terminology: The text uses multiple terms to refer to the BP monitoring system, including "wearable system," "wireless system," and "flexible, lightweight wristband." Using consistent terminology can help readers follow the description more easily.

Our response: We thank the referee for the valuable suggestion. We uniformed the terminology throughout the paper.

Modifications: In Line 29, manuscript, we modified the text as “Here, we report a thin, soft, miniaturized system (TSMS) that combines a conformal piezoelectric sensor array, an active pressure adaption unit, a signal processing module, and an advanced machine learning method, to allow real wearable, continuous wireless monitoring of ambulatory artery BP.”

In Line 84, manuscript, we modified the text as “Here, we report a class of materials, devices, mechanic designs, data processing methods and integration strategy for a thin, soft, miniaturized system (TSMS), that is capable for continuously monitoring of BP in real wearable format, with whose accuracy is comparable to the professional medical equipment. The TSMS composes three subsystems, including a sensing module to detect the blood pulse wave, an active pressure adaption module to provide powerful back pressure for improved interfacial properties, and a data processing module to in-situ process and transmit the measured data to a graphical user interface (GUI).”

In Line 106, we modified the text as “**Fig. 1a** shows the schematic illustration of the TSMS in a wearable wristband format, where the overall dimension is 15 cm in length, 35 mm in width and 4 mm in thickness. The TSMS includes three subsystems of a piezoelectric based pulse sensing system, an active pressure adaptation system and a data sampling/transmitting system (**Fig. 1b, Supplementary Fig. 2**).”

In Line 153, manuscript, we modified the text as “The low power consumption of the TSMS keeps it from accumulating heat in the regional area.”

In Line 156, manuscript, we modified the text as “To our best knowledge, the TSMS in this work providing a totally wearable, user-friendly interface is the most advanced continuous BP monitoring device in terms of wearability, accuracy, continuity, dynamics, and many other detailed aspects (**Fig. 1g, Supplementary Table 1**).”

In Line 161, manuscript, we modified the text as “The TSMS encapsulated by silicone in a wristband format provides an easy and efficient way for users to wear, where the sensor array can detect the mechanical pulse caused by blood propagation.”

In Line 361, manuscript, we modified the heading and text as “**Statistical and dynamic evaluation of the TSMS for wireless continuous and dynamic BP monitoring.**”

To validate the measure accuracy of the TSMS system, a total number of 87 volunteers were selected to continuously monitor their BP for 2 minutes with BP measured by commercial noninvasive arterial blood pressure monitor (CNAP) and commercial sphygmomanometer (Omron HEM-7156T) as references. **Extended Data Fig. 9a** shows the statistic error distribution of the 87 volunteers, where most of error bars are less than 10 mmHg regardless of SBP or DBP. Moreover, quantitative statistics on SBP and DBP show that excellent measurement accuracy was achieved with the TSMS, with over 70% for error < 5 mmHg and over 98% for error < 15 mmHg (**Extended Data Fig. 9b, 9c, Supplementary Fig. 20**). Moreover, to validate the measurement robustness of the TSMS, dynamic evaluation process was conducted. The evaluation is associated with wearing TSMS on the right wrist to continuously measure BP, and at the meantime commercial equipment (BioPAC) equipped with CNAP monitor, which has been proved providing clinically acceptable accuracy⁶⁵, attached on the same volunteers to measure reference BP, as illustrated in **Fig. 4a** and **Supplementary Video 2**. **Fig. 4b** shows the comparison between BP patterns measured by the TSMS and the BioPAC of a volunteer, in which up to 25 minutes of continuous BP data is measured by the TSMS and agrees very well with those measured by BioPAC, indicating the excellent performance of the TSMS. Besides, the statistical comparison between the TSMS and BioPAC also shows excellent agreement, especially in DBP, with the same mean value as well as a slightly smaller dynamic range in 25 minutes (**Fig. 4c, Supplementary Fig. 21**). To further study the capability of our TSMS in dynamic monitoring of BP, hand grip and cold pressor process (HGCP) cycles, which has been proved an efficient way to increase BP due to the triggered sympathetic response^{8,66}, was adopted to realize a large dynamic range of BP (**Fig. 4d**).”

In Line 387, manuscript, we modified the text as “The continuous BP patterns measured by the TSMS show satisfying agreement, with the predicted maximum SBP and DBP at 158 mmHg and 109 mmHg, respectively.”

In Line 399, manuscript, we modified the text as “Meanwhile, the statistic BP measured with a commercial cuff-based sphygmomanometer is regarded as the reference BP to evaluate the BP measured by the TSMS.”

In Line 414, manuscript, we modified the text as “To further explore the significance of continuous BP monitoring by the TSMS, we tracked the continuous BP data of the borderline hypertension individual and hypertension individual for a week (**Extended Data Fig. 10**)”

In Line 435, manuscript, we modified the text as “One typical application of the TSMS would be providing medical instructions for patients with sustained hypertension.”

In Line 454, manuscript, we modified the text as “Excellent measurement robustness and accuracy, with 0.002 ± 3.335 mmHg for DBP and 0.025 ± 4.582 mmHg for SBP, was achieved by the TSMS.”

In Line 461, manuscript, we modified the text as “Benchtop studies, structural design, theoretical simulation, and initial trials on a total number of 87 volunteers have demonstrated the feasibility and practical utility of the TSMS in continuous BP monitoring. Furthermore, dynamic BP tracking of a hypertension individual and a borderline hypertension individual reveals the BP changes during a day and a week. We envision that the TSMS could play a crucial role in realizing precise BP control for hypertension individuals and cardiovascular disease prevention through continuous BP monitoring and the development of personalized diagnosis and treatment schemes.”

Comment 6: Supplementary Fig. 6. In Fig 6 (c), It appears that the system's response time is faster when the one-way valve is not assembled compared to when it is. Please explain.

Our response: We thank the referee for the useful suggestion. There are two reasons causing the slower response when the one-way valve is equipped. On the one hand, the diameter of the inner hole is smaller than that of the silicone hose, which will introduce damping to stabilize the system. On the other hand, the air flow in the high pressure zone will deform the silicone cover to generate the air gap, which will further increase the system damping. In summary, the response time is increased due to the presence of extra system damping when the one-way valve is equipped. It is worth noting that although the response time increases when the one-way valve is equipped, the maximum pressure (40 KPa) can be effectively applied within 500 ms (Supplementary Fig. 6). In contrast, increased damping will greatly enhance the system stability and robustness. We added an explanation on the influence of the one-way valve on response time and system stability.

Modifications: In Line 170, manuscript, we modified the text as “The pressure up to 40 kPa provided by the active pressure adaptation system guarantees the continuous and accurate monitoring of piezo response generated by blood pulse (**Extended Data Fig. 2b-d**). The one-way valve, consisting of a perforated silicone hose, an isolation film, and a silicone cover, can effectively maintain the pressure in the airbags when the pump is turned off (**Extended Data Fig. 3a**). When the pump is turned on, there will be gaps between the bonding sites on the silicone cover and the silicone hose due to the pressure difference across the silicone cover (**Extended Data Fig. 3b**). In contrast, when the pump is off, the silicone cover will be pressed tightly against the silicone hose, maintaining a high pressure in the airbag side. With the one-way valve, the pressure in the airbags can maintain at a high level for a long time, with pressure decreasing to 30 % of the original pressure after 1 hour (**Extended Data Fig. 3c-g**). It is worth noting that, although the pressure response time increases with the one-way valve equipped owing to increased system damping, the pressure (up to 40 KPa) can still be applied rapidly in less than 500 ms (**Supplementary Fig. 6**). While the increased system damping greatly enhances system stability, with less pressure fluctuations.”

Comment 7: Line 162: it could benefit from more specific details on the materials and components used. For instance, it would be helpful to know the specific type of PZT used for sensors, as well as the acoustics performance of the sensors.

Our response: We thank the referee for these valuable suggestions. We added material details of the piezoelectric pressure sensor and added explanation on material selection. For the acoustics performance of the PZT, in this work, we focused on the sensing performance, with relative low working frequency compared to acoustic applications. Therefore, we did not conduct experiments to study the acoustics performance of the sensors.

Modifications: In Line 92, manuscript, we added explanation on the material selection. “Highly chemical/physical stability of the piezoelectric material PZT 5H enables the sensor array with excellent uniformity and robustness while maintaining superior sensitivity due to its high piezoelectric coupling coefficient⁴⁰.”

In Line 163, manuscript, we added details on material type and modified the text as “While the use of PZT 5H as sensors is based on its excellent electrical, physical and chemical stabilities (**Extended Data Fig. 4**), which also leads to interfacial contact issues with the skin due to its mechanical stiffness.”

Comment 8: Line 183-184, “the change of pressure in the airbag during 5 pumping phases, where the pumping duration of the pump was set at 35 ms for all five phases,” it is unclear whether the five phases have the same interval or different intervals. A more detailed explanation of the pumping phases and their associated intervals would greatly enhance the understanding of the experimental setup.

Our response: We thank the referee for the useful suggestion. We added explanation on the experimental setup during multi pumping phases. Specifically, we used the same pumping duration and interval, with 35 ms pumping and 3500 ms interval, for all the five pumping phases.

Modifications: In Line 189, manuscript, we added explanation on experimental setup and modified the text as “**Fig. 2b** shows the change of pressure in the airbag during 5 pumping phases, where the pumping duration and pumping interval of the pump was set at 35 ms and 3500 ms for all five phases, respectively.”

Comment 9: Line 185: It is unclear what is meant by "less than the systolic BP" in the sentence "The pressure increases from 0 to 12 kPa, corresponding to 0-90 mmHg, which is less than the systolic BP," It would be helpful to explain this further or provide a reference.

Our response: We thank the referee for the valuable suggestion. As we know, an increased backpressure contributes to better signal quality due to the powerful support

provided by the micro-airbag. However, the artery will be closed with an external loading pressure over its inner pressure provided by blood propagation, which will cause discomfort to users and thus is not suitable for long-term continuous measurement. Therefore, we converted the pressure from mmHg to kPa to determine the maximum pressure that is allowed for the micro-airbag without discomfort generated while maintaining good signal quality. In practice, the maximum pressure 12 kPa, corresponding to 90 mmHg BP, will not cause long-term close of artery, and further discomfort. Consequently, we set the maximum pressure of the pressure adaptation module at 12 kPa. We added an explanation on how the maximum pressure was determined in manuscript to facilitate understanding.

Modifications: In Line 185, manuscript, we added an explanation on the maximum pressure “The pressure adaptation module serves a powerful support to the sensor for increasing the sensing signal quality. However, too high back pressure may also cause discomfort to users due to the closure of artery. Therefore, we set the maximum pressure at 12 kPa (90 mmHg) to avoid uncomfortable feelings while maintaining satisfactory signal quality (**Supplementary Fig. 7**)”

Comment 10: Line 189-191: Consider adding some contextual information for the reader to better understand the significance of the results presented about the close-looped feedback strategy. For example, why do we need closed-loop control instead of open loop control here, is there any disturbance? If so, is the control strategy robust enough to against it? It would be helpful to explain this further or provide a reference.

Our response: We thank the referee for the insightful comment. Normally, pressure sensor based continuous BP monitoring devices suffer from poor interfacial performance as the difference in intensity of blood pulse across individuals. For instance, a weaker pulse beat is normally felt on skin in high BMI individual groups, which will reduce signal quality, even not processable with wearable circuits. Therefore, we developed the active pressure adaptation module to intelligently adjust the applied back pressure for satisfactory signal quality. For the close-looped feedback, we need to detect if available signal is measured to determine if the applied pressure is enough or a higher pressure is needed for BP estimation. Besides, higher pressure might generate discomfort to users due to the close contact between the sensor and skin. By taking these factors into consideration, we proposed the multi pumping control strategy for the active pressure adaptation module for the realization of great measurement accuracy while maintaining excellent user experience. We added an explanation on why a close-looped control strategy is required for the pressure adaptation module. Furthermore, to evaluate if the close-looped control strategy is robust enough to against disturbance, we conducted experiments to study how the close-looped control strategy improves measurement accuracy while maintaining user comfort. **Supplementary Fig. 7** presents how the close-looped control strategy improve the signal quality, when signal distortion is detected (highlighted in red dotted rectangular), an extra pumping phase is performed to inflate the micro airbag to improve the signal quality. From

Supplementary Fig. 7, it is obvious that the signal quality is greatly improved after extra pumping phase was applied. We added explanation on how the close-looped control strategy improve the measurement accuracy in the manuscript.

Modifications: In Line 196, manuscript, we modified the text as “It is worth mentioning that we developed a close-looped feedback strategy between the MCU and the mobile GUI to optimize the pressure in the airbag by real time adjusting/controlling the pump for optimizing signal quality. Specifically, the gradually decreasing pressure in the micro airbag body movement would introduce fluctuations and distortions to the measured pulse wave. When distortion is detected, the pump will inflate the micro airbag to improve the signal quality (**Supplementary Fig. 8**).”

Supplementary Fig. 8. Robustness evaluation of the close-looped control strategy against signal distortion.

Comment 11: Line 199: It would be useful to clarify what is meant by "different situations." Are the situations referring to different wrist elbow joint angles?

Our response: We thank the referee for the useful suggestion. Yes, different situations refer to different wrist/elbow joint angles. We modified different situations to different joint angles to avoid misunderstanding.

Modifications: In Line 208, manuscript, we modified the text as “With the assistance of the active pressure adaptation system for providing sufficient backpressure to the sensors, the piezo response generated from blood propagation can be effectively detected under wrist/elbow bending at different angles.”

Comment 12: Line 214-215: it would be useful to clarify what is meant by "physical features" and "time information" that are extracted from the original pulse waveform.

Our response: We thank the referee for the useful suggestion and appreciate your comment on the reference of "physical features" and "time information". We agree these two words may bring ambiguity and hence we have modified them on the manuscript. The "time information" referred to the values in x axis (time axis) of the extracted feature points in waveform such as the feature 1-5 and 7-18 (**Supplementary Table 2**). And the "physical features" refer to other features extracted from waveform morphology and the features derived from the waveform, including Systolic peak height, Diastolic peak height, Relative augmentation index and Inflection point area ratio (feature 19-22).

Modifications: In line 287, manuscript, we added detailed description on physical features and time information and modified the text as "Time-domain characteristics (the x index of the extracted feature points) and morphological features (features extracted from waveform morphology and the features derived from the waveform) such as Systolic peak height, Diastolic peak height, Relative augmentation index and Inflection point area ratio."

Comment 13: Line 257-259: it would be helpful to provide a citation or reference for the claim that respiration and body movements could lead to distortion of the signal morphology.

Our response: We thank the referee for the useful suggestion. We added two references related to the signal distortion caused by respiration and body movements.

Modifications: In Line 270, we added a reference and modified the text as "It is possible to get blood pressure values from the raw pulse waveform, but respiration and body movements could lead to distortion of the signal morphology, thus affecting the predicted accuracy^{55,56}."

Newly added references

55. Shin, H. S., Lee, C. & Lee, M. Adaptive threshold method for the peak detection of photoplethysmographic waveform. *Computers in Biology and Medicine* 39, 1145–1152 (2009).
56. Chuang, C.-T., Chang, T., Chiang, Y.-T. & Chang, F.-R. Heart Rate Monitoring Using a Slow–Fast Adaptive Comb Filter to Eliminate Motion Artifacts. *J. Med. Biol. Eng.* 36, 833–842 (2016).

Comment 14: Line 265: it is recommended to clarify what is meant by "relevant standards" and provide a reference or citation.

Our response: We thank the referee for the valuable suggestion. Organizations like Association for the Advancement of Medical Instrumentation/European Society of Hypertension/ International Organization for Standardization (AAMI/ESH/ISO) and British Hypertension Society (BHS) have proposed relevant standards for performance evaluation of continuous BP monitoring devices. Specifically, these standards put requirement on sample size and experimental settings. We added references of these standards in the manuscript to facilitate better understanding.

Modifications: In Line 277, manuscript, we added these standards as references and modified the text as “Many works have been proposed with PTT-based single indicator methods to estimate BP⁵⁹⁻⁶¹, but they could not satisfy the relevant standards^{62,63} in sample size and require extra calibrations owing to insufficient features for model learning.”

Newly added references

62. Stergiou, G. S. et al. A Universal Standard for the Validation of Blood Pressure Measuring Devices. *Hypertension* 71, 368–374 (2018).
63. O’Brien, E., Atkins, N., Mee, F. & O’Malley, K. Evaluation of Blood Pressure Measuring Devices. *Clinical and Experimental Hypertension* 15, 1087–1097 (1993).

Comment 15: Line 281: Provide more information about the selection criteria for the volunteers to help readers understand the representativeness of the sample.

Our response: We thank the referee for the comprehensive suggestion. We add an explanation on the volunteer selection criteria in the manuscript.

Modifications: In Line 295, manuscript, we modified the text as “17 volunteers covering a wide range of age and BMI were selected to study the BP variations during the continuous BP measurement (detail process can be found in method)”

Comment 16: Line 296: Consider clarifying what is meant by "limited sample size". How many samples were included in the study, and why is this number considered to be limited? Providing this information will help the reader understand the significance of the study findings.

Our response: We thank the referee for these insightful comments. In this work, we included data from 17 subjects for model training and 87 people for data validation, which can be considered as a small group, and may limit the generalizability of our model. For example, many studies for BP estimation often utilized the public database such as the MIMIC database which contains data from thousands of ICU patients. As we report a completely new system consists of both hardware and BP estimation model, so we used the data recorded from our own devices. To address this limitation, we

applied data augmentation techniques to increase the training set to improve the learning ability of model and avoid the overfitting risk.

Modifications: In line 311, we added explanation on data size and modified the text as “Our data was collected from the participants who used our devices, which differs from the large-scale public databases that are often derived from controlled environments such as hospitals. Therefore, our model is suitable for the users in daily life scenarios.”

Comment 17: Line 297: Consider including a brief description of how the XGBoost algorithm works rather than just in Fig 3b. This will help readers who are not familiar with the algorithm to understand why it was selected as the base model.

Our response: We thank the referee for the constructive feedback and valuable suggestions. XGBoost is a popular framework for gradient boosting, which is a type of ensemble learning in machine learning. The name XGBoost stands for Extreme Gradient Boosting, and it is an optimized implementation of decision trees. When training the model, the algorithm builds a decision tree that captures residual errors from the previous tree. The model combines the predictions made by multiple simpler models to produce a final strong model. We considered several key factors, including calculation speed, prediction accuracy, and model interpretability, when selecting XGBoost as the base model. Compared to the neural network, based on the feature engineering it provides us clear insights into features importance and how they impact the predictions. Additionally, XGBoost operates faster than deep neural networks. Furthermore, the testing accuracy of XGBoost has demonstrated better performance than common machine methods such as MLP, K-Nearest Neighbor (KNN), Support Vector Regression (SVR), and Adaptive Boosting (Adaboost) algorithm using our dataset (**Figure 3d, Supplementary Fig. 12**).

We appreciate your suggestion, and it is helpful for strengthen the paper. Therefore, we have revised the text to make it more precise and clearer.

Modifications: In line 313, manuscript, we modified the text as “Considering possible overfitting risk of large models such as deep neural networks, a simple model may achieve better performance. Ensemble learning is a suitable candidate that can reduce the risk of overfitting. Extreme Gradient Boosting (XGBoost, **Fig. 3b**) is an extension of the Gradient Boosting Trees algorithm that combines the predictions made by multiple simpler models to produce a final strong model. It performs better than a single learner and takes less computational time than a complex network such as deep neural network. Thus, it was selected as the base model for BP estimation.”

Comment 18: Line 329: Provide more context for the comparison with BioPAC: The paragraph compares the BP patterns measured by the wireless wristband with those measured by BioPAC, but it doesn't explain what the accuracy of products BioPAC is or why it's being used as a reference.

Our response: We thank the referee for the comprehensive comment. In fact, BioPAC system equipped with CNAP monitor has been proved a reliable candidate for continuous BP measurement while providing clinically acceptable accuracy compared to the gold standard for continuous BP measurement (invasive arterial cannula). We added an explanation on the accuracy of CNAP system and added a reference related to its high measurement accuracy to facilitate better understanding.

Modifications: In Line 371, manuscript, we added explanation on the measurement accuracy of CNAP system and modified the text as “The evaluation is associated with wearing TSMS on the right wrist to continuously measure BP, and at the meantime commercial equipment (BioPAC) equipped with CNAP monitor, which has been proved providing clinically acceptable accuracy⁶⁵, attached on the same volunteers to measure reference BP, as illustrated in **Fig. 4a and Supplementary Video 2.**”

Newly added references

65. Dewhurst, E. et al. Accuracy of the CNAP monitor, a noninvasive continuous blood pressure device, in providing beat-to-beat blood pressure readings in the prone position. *Journal of Clinical Anesthesia* 25, 309–313 (2013).

Comment 19: Line 356: It would be helpful to explain that the cuff-based sphygmomanometer is considered the gold standard for BP measurement and was used as a reference to compare the accuracy of the wristband readings.

Our response: We thank the referee for the important suggestion. We added the description of cuff-based sphygmomanometer as the gold standard for BP measurement in the manuscript.

Modifications: In Line, 399, manuscript, we modified the text as “Meanwhile, the statistic BP measured with a commercial cuff-based sphygmomanometer, which is the gold standard for routine BP measurement, is regarded as the reference BP to evaluate the BP measured by the TSMS.”

Comment 20: Line 394 and 404: Please use more precise language here. It contains some imprecise language that could be clarified to improve its professional tone. The phrase "may cause heart and kidney damage due to sharp fluctuation of BP" could be revised to more accurately reflect the relationship between fluctuating BP and organ damage. Similarly, the phrase "unreliable interfacial contact between the device and artery" could be revised to more specifically describe the issues with signal quality and measurement accuracy.

Our response: We thank the referee for these useful suggestions. We revised these descriptions to precise language to reflect the significance of continuous BP measurement and the performance improvement of our design strategy.

Modifications: In Line 438, manuscript, we modified the text as “However, the antihypertensive drug might not control the BP effectively throughout the entire day and night, during which heart and kidney damage might be caused by the sharp BP fluctuation.”

In Line 448, manuscript, we modified the text as “Moreover, the unreliable interfacial contact between the device and artery will introduce extra noise signals to pulse wave, such as sliding noise caused by loose contact and occasional error, which will greatly reduce signal quality, and thus leading to poor measurement accuracy.”

Comment 21: Line 424: Consider providing more information on the limitations of the study, such as the small sample size or the potential confounding variables. This context would help the reader understand the study's limitations and potential biases.

Our response: We thank the referee for the comprehensive suggestion. We added description on the limitations of this work to facilitate better understanding.

Modifications: In Line 468, manuscript, we added explanation on the study and modified the text as “To realize long-term accuracy monitoring of continuous BP pattern, comprehensive pre-train process to the data model is needed with a balanced distribution on BP range and other factors that contribute to BP fluctuations. This study, instead, focused on limited size of human participants actual scenes. We believe further expanding the number of human participants and improving BP distribution will improve the BP estimation accuracy.”

Grammatical aspects

Comment 22: Line 48: The sentence "..., heart rate, cardiac output, which are closely relate to CVDs" should be revised to "..., heart rate, and cardiac output, which are closely related to CVDs.”

Our response: We thank the referee for the useful suggestion. We addressed this.

Modifications: In Line 48, manuscript, we revised the text to “Blood pressure (BP) as an important biomarker provides remarkable insights into many hemodynamic parameters, such as stroke volume, heart rate, and cardiac output, which are closely relate to CVDs³⁻⁵.”

Comment 23: Line 97-100: The sentence "Compared to other cuffless BP monitoring devices, ..., cuffless blood pressure monitoring devices" should be divided into two sentences like: "Compared to other cuffless BP monitoring devices, the integrated system reported here shows great advantages in terms of wearability, continuance, dynamics, and measurement accuracy. It is characterized as Grade A based on the

standard for wearable, cuffless blood pressure monitoring devices.” The overall structure of the main could be improved by breaking down longer sentences into shorter ones and avoiding the use of passive voice wherever possible.

Our response: We thank the referee for the useful suggestion. We modified the description of the advantages of our work over published works.

Modifications: In Line 100, manuscript, we modified the text as “Compared to other cuffless BP monitoring devices (Supplementary Table 1), this work addresses the issues in system integration level, interfacial performance, and BP estimation model, thus showing great advantages in terms of wearability, continuance, dynamics, and most importantly, measurement accuracy (Grade A level).”

Comment 24: Line 156, In the sentence "To our best knowledge,...,wearability, accuracy, continuance, dynamics, ...," "continuance" seems like an unusual word choice. It could be revised to durability.

Our response: We thank the referee for the insightful suggestion. We used continuance to reflect if BP is measured continuously. To avoid the confusion by the word “continuance”, we revised it to continuity.

Modifications: We modified the text and Fig. 1g. In Line 156, manuscript, we modified the text as “To our best knowledge, the TSMS in this work providing a totally wearable, user-friendly interface is the most advanced continuous BP monitoring device in terms of wearability, accuracy, continuity, dynamics, and many other detailed aspects (**Fig. 1g, Supplementary Table 1**).”

Figure 1. Working principle and layouts of the wearable wireless continuous blood pressure monitoring system. (a). Schematic diagram of signal conversion from piezo response to continuous blood pressure that is presented in a mobile GUI. Physical distance between two sampling sites and time difference in two sensing units were utilized to calculate localized PWV. Pulse wave features, together with localized PWV were transmitted to data for the estimation of beat-to-beat BP. (b). Explosive view of the wireless wristband, with three subsystems, sensing module, force generation module and signal processing module. (c). Optical image of the wireless wristband worn on user's wrist joint. (d). Optical image of all the system components before sealed in the silicone wristband. (e) and (f). Optical images of the sensor array and micro airbag array suffering from mechanical deformations. (g). Technical comparison between our device and published works for continuous BP monitoring in terms of wearability, accuracy, dynamics, continuance and wireless.

Comment 25: Line 162-164: The sentence "While the use of PZT as sensors based ... stiffness," it would be clearer to write "While the use of PZT as sensors is based on its excellent electrical, physical and chemical stabilities, it also leads to interfacial contact issues with the skin due to its mechanical stiffness."

Our response: We thank the referee for the useful suggestion. We revised the

manuscript as suggested.

Modifications: In Line 163, manuscript, we modified the text as “While the use of PZT 5H as sensors is based on its excellent electrical, physical and chemical stabilities (Extended Data Fig. 4), which also leads to interfacial contact issues with the skin due to its mechanical stiffness.”

Comment 26: Line 165 and 166: there are a few instances where plural and singular forms are mixed, such as "micro-airbag" and "micro pump" in the same sentence. It would be clearer to use consistent grammar throughout.

Our response: We thank the referee for the important suggestion. We revised the manuscript throughout to use consistent grammar expression with “micro airbag” and “micro pump”.

Modifications: In Line 165, manuscript, we modified the text as “Therefore, we designed an active pressure adaptation system, consisting of a pair of silicone based micro airbag (20 mm × 30 mm × 1.5 mm), a micro pump (19 mm × 21 mm × 3.6 mm) and a one-way valve, that can provide extra backpressure to the sensor array and thus maintain good contact behavior between the sensor and skin/artery.”

Comment 27: Line 194: 1. The sentence "The build-in counter in ... for relocating to users via the mobile GUI" has a grammatical error. It could be rephrased as "The built-in counter in the MCU counts the pumping phases and inflates the airbag, allowing for a maximum of 5 pumping phases corresponding to a peak pressure of 12 kPa (Fig. 2b) before sending warning information to users via the mobile GUI to relocate."

Our response: We thank the referee for the useful suggestion. We addressed the grammatic error correspondingly.

Modifications: In Line 204, manuscript, we modified the text as “The build-in counter in the MCU counts the pumping phases and inflating the airbag, allowing for a maximum of 5 pumping phases corresponding to a peak pressure of 12 kPa (Fig. 2b) before sending warning information for relocating to users via the mobile GUI.”

Comment 28: Line 248: Check for typo: “senor array...”, should be “sensor array”

Our response: We thank the referee for the important comments. We check for typo throughout the manuscript and modified them accordingly.

Modifications: In Line 258, manuscript, we modified the text as “In this strategy, the piezo responses in the sensor array are sampled by the ADC with a high sampling frequency at 4000 Hz (2000 Hz for each sensing unit), and then calculated in the MCU to extract Δt before resampling with a low frequency at 200 Hz (Supplementary Fig.

11)''

REVIEWER COMMENTS

Reviewer #1 (Remarks to the Author):

The author presents a wearable system for continuous wireless monitoring of artery blood pressure including a piezoelectric sensor array, pressure adaptation unit, a signal processing module and a machine learning method. The author has addressed some concerns of reviewers, but the reviewer cannot fully agree on remaining concerns. For these reasons, the reviewer cannot recommend this manuscript for publication in the Nature Communications.

1. Author's piezoelectric sensor does not have novelty compared to previous piezoelectric blood pressure sensors of "Adv. Mater., 34, 2110291, 2022", which uses a few hundred micrometers PZT film for measuring the waveform of similar. The difference is the addition of active pressure adaptation and blood pressure inference. However, even with these additions, it's hard to see that the blood pressure waveform signal was clearly measured from Extended Data Figure 8. There was no variation in the actual blood pressure inference depending on the pressure, along with significant movement noise.

2. There are recent publications in this field based on more advanced piezoelectric sensor, such as Nature Biotechnology DOI: 10.1038/s41587-023-01800-0 and Adv. Mater. DOI: 10.1002/adma.202301627. It is questionable whether this manuscript provide sufficient novelty and scientific contribution in addition to the above references.

3. Serious issue is that the revised manuscript claimed that total of 87 volunteers for blood pressure measurement, adding them in Extended Data. However, the main MS described only the accuracies of blood pressures in the initial 17 participants of previous version.

Then why did the author add 87 volunteers?

Current manuscript may lead readers to misunderstand the measurement condition.

Reviewer #2 (Remarks to the Author):

The authors have addressed all questions and it can be accepted.

Reviewer #3 (Remarks to the Author):

As a system the reported work is very impressive. Most of previous review comments were very well responded. One more comment, authors may conduct a thorough survey on wearable blood pressure monitoring techniques including wearable ultrasound for blood pressure monitoring, etc.

Response to comments of Referee #1

Comments from Referee #1:

Summary Comment: The author presents a wearable system for continuous wireless monitoring of artery blood pressure including a piezoelectric sensor array, pressure adaptation unit, a signal processing module and a machine learning method. The author has addressed some concerns of reviewers, but the reviewer cannot fully agree on remaining concerns. For these reasons, the reviewer cannot recommend this manuscript for publication in the Nature Communications.

Our response: We thank the referee for these valuable comments. We carefully reviewed these comments to address these issues point by point and revised the manuscript accordingly.

Modifications: None.

Common 1: Author's piezoelectric sensor does not have novelty compared to previous piezoelectric blood pressure sensors of "Adv. Mater., 34, 2110291, 2022", which uses a few hundred micrometers PZT film for measuring the waveform of similar. The difference is the addition of active pressure adaptation and blood pressure inference. However, even with these additions, it's hard to see that the blood pressure waveform signal was clearly measured from Extended Data Figure 8. There was no variation in the actual blood pressure inference depending on the pressure, along with significant movement noise.

Our response: We thank the referee for understanding the technical details, while the novelty of our work is not neither sensor nor pressure waveform measurement. We present a class of materials, electronics, device design and system integration to realize real time blood pressure monitoring, which has been recognized as a very challenge point in the community. So, our work is completely difference from the piezoelectric blood pressure sensor of "Adv. Mater., 34, 2110291, 2022", as our system addresses many engineering challenges from interfacial properties between skin and sensors, system level integration/combination of materials, devices with practical electronics. In fact, we've summarized the merits of our system over the reference (Adv. Mater., 34, 2110291, 2022) in detail in the first version of reversion letter, and we also list them here for your reference.

For the active pressure adaption module, in **Extended Fig. 8 a-d**, we study the influence of different back pressure levels on BP estimation results, where it is obvious that the back pressure level in the micro-airbag contributes only to the signal amplitudes, while maintaining the BP estimation accuracy. In **Extended Fig. 8 e-f**, we aimed to study how the active pressure adaption module improves the interfacial performance of our system compared to that without back pressure equipped. The results in previous version of

manuscript might be not convincing enough, we further conducted experiments to study how the active pressure adaption module improves signal quality against body movement and present the results in **Extended Fig. 9**. **Extended Fig. 9** lists the results of measured pulse signals comparison with and without backpressure applied under a set of forearm movement, that is wrist joint rotation, forearm raising and violent forearm shaking. From the results presented in **Extended Fig. 9**, it is obvious that the signal distortion is significantly improved with the active pressure adaption module, where pulse features are well maintained even under severe forearm shaking (inset of **Extended Fig. 9f**). From these results, we believe that it is convincing enough to prove the great improvement of our active pressure adaption module on interfacial stability.

Merits of our system over the reference (Adv. Mater., 34, 2110291, 2022):

From the points of device design and system integration, we presented a highly integrated system, with sensor array, back pressure adaption module, and signal processing module integrated in a totally wearable format. In comparison, the reference (Adv. Mater., 34, 2110291, 2022) only presented a wearable sensor, fixed on skin with the assistance of tapes. The rigid signal sampling module was not integrated with sensors, and thus is not suitable for wearable measurement.

From the point of device stability and adaptability, the sensor in the reference was roughly fixed with a tape, where the interfacial stability needs to be addressed for long-term monitoring. In addition, encapsulation will further increase the interfacial instability, which is also a common issue in skin-integrated electronics. Based on this consideration, we presented a universal strategy, that is the active pressure adaption module to address the interfacial performance of skin-integrated electronics.

From the point of BP estimation performance, the BP calibration method in the reference relied on the calibration factor between the measured pressure wave and continuous BP wave, where repeated calibration process is required before each measurement. In comparison, our pretrained model provide a better BP estimation performance without repeated calibration process required.

Modifications: In Line 347, manuscript, we added description of how the active pressure adaption module improve signal quality against a set of forearm movement and modified the text as “**Extended Data Fig. 9** shows the comparison in measured pulse wave with and without backpressure applied under the situations of wrist joint rotation, forearm raising and forearm shaking, where it can be found obvious signal distortion happened for the device without backpressure. In comparison, there is only slightly fluctuations in signal amplitude to a set of body movement and no obvious signal distortion when backpressure is applied. It can be concluded that the introduction of the active pressure adaption module significantly improves the system robustness against body movement while maintaining measurement accuracy.”

Extended Data Figure 9. Performance validation of the active pressure adaptation module on interfacial stability enhancement. (a) and (b). Comparison on measured pulse signal with and without backpressure applied under 90° wrist rotation. (c) and (d). Comparison on measured pulse signal with and without backpressure applied under forearm raising. (e) and (f). Comparison on measured pulse signal with and without backpressure applied under severe forearm shaking.

Common 2: There are recent publications in this field based on more advanced piezoelectric sensor, such as Nature Biotechnology DOI: 10.1038/s41587-023-01800-0 and Adv. Mater. DOI: 10.1002/adma.202301627. It is questionable whether this manuscript provide sufficient novelty and scientific contribution in addition to the above references.

Our response: We thank the referee for pointing out the two recent papers, which we've also carefully read. However, these two papers are difference from our work. Especially for the work published in Nat. Nano, which adopted an ultrasound patch to

detect the variation of arterial diameter, showing many drawbacks in continuous BP monitoring in calibration process, interfacial performance, and system power consumption. We summarized the merits and contributions of our system over them in continuous BP measurement.

For the first paper (Nature Biotechnology DOI: 10.1038/s41587-023-01800-0), the authors developed a wireless ultrasound system to detect the variations of artery diameter during blood circulation. Then the diameter variation of targeted artery is converted to continuous BP wave based on a theoretical model:

$$p(t) = p_d * e^{\beta \left(\frac{D(t)}{D_d} - 1 \right)}$$

and

$$\beta = \frac{D_d \ln(p_s / p_d)}{D_s - D_d}$$

D_s and D_d stand for the arterial diameters at systolic BP (p_s) and diastolic BP (p_d), respectively. From this system, we summarized the merits of our system over it in continuous BP measurement and listed these points as follows:

From the point of BP calibration process, the reference (Nature Biotechnology DOI: 10.1038/s41587-023-01800-0) relies on the aforementioned model to calculate continuous BP from diameter variation in targeted artery. However, isolated systolic BP and diastolic BP need to be involved in the calculation of vascular stiffness related coefficient β , indicating that personalized calibration process and well-trained professionals must be involved for each user to calibrate the device, which significantly increase the usage barrier. Moreover, for users suffering from vascular diseases, the arterial stiffness varies along the artery tree, which will increase the calibration error of $\beta^{2,3}$, and thus reduce the BP measurement accuracy. On the contrary, there is no personalized calibration process required in our machine learning based data model. Moreover, we extract pulse features and localized PWV for BP estimation, which takes the local stiffness property of artery into consideration, thus showing wider applicability over the reference.

From the point of interfacial stability, the reference encapsulated the ultrasonic probe in silicone elastomer and mounted the system on skin with the assistance of medical silicone adhesives. As we all know, the ultrasound system puts high requirements on the interfacial acoustic impedance, as the mismatch in interface acoustic impedance will significantly increase the interface reflection, thereby reducing the echo signal quality. That is why even commercial bulky ultrasound systems need to use contrast agent to increase the echo signal quality. In the reference, there will be air gaps generated between the ultrasound probe and skin during long-term wearing, which will greatly reduce the signal quality, and thus bring errors for BP calculation. In comparison, our system provides a user-friendly interface without any adhesive involved, and the active pressure adaption module provides sufficient support to the pressure sensor array, enabling the long-term stable measurement of continuous BP.

From the point of power consumption, for wearable system, portability is a key factor that need to be considered while designing it. At present, most of wearable systems are designed and optimized towards high power efficiency to increase their lifetime while maintaining wearability. The ultrasound probe in the reference needs high frequency (over 2 MHz) square or sine wave to power, which will greatly increase the power consumption of the circuit board due to the high frequency signal generator and high sampling rate of the ADC. It is tested in the reference that a commercial lithium-ion battery with a capacity of 2000 mAh, occupying over 20 cm³ of space, can only power the system for 12 h. In comparison, our system provides a power saving strategy for continuous BP measurement, with an 80 mAh lithium-ion battery powering the system for almost 2 days.

For the second paper (Adv. Mater. DOI: 10.1002/adma.202301627), authors adopted similar piezoelectric sensor that embedded in a wristwatch to detect continuous pulse wave. Then authors converted the continuous pulse wave into BP wave via a transfer factor, similar to that utilized in previous work (Adv. Mater., 34, 2110291, 2022), which will definitely arise calibration issue. Moreover, the interfacial stability and wearability could be further improved compared to that proposed in our system. We summarized the advantages of our system over it and listed them as follows:

From the point of BP calibration process, the reference (Adv. Mater. DOI: 10.1002/adma.202301627) adopted a transfer function to convert the measured continuous pulse wave into continuous BP. Specifically, the authors recorded the peak and trough points continuously for 30 s, while measuring an isolated SBP and DBP with cuff-based tonometry. Then the authors mapped average peak and trough voltage values during 30 s as SBP and DBP, respectively via a conversion factor. Obviously, personalized calibration process is required for each user as different mechanical properties of skin and arterial wall contribute to measured peak and trough values. Most importantly, different back pressure levels affect the output voltage of the pressure sensor in response to blood propagation. A higher back pressure level will increase both the peak voltage and the trough voltage of the measured signal, which means users must ensure that the same pressure level is applied to the pressure sensor after each wearing or the device needs to be recalibrated. As we all know, it is impossible to maintain the same backpressure provided by the watch strap during two times of wearing, which indicates that the device in the reference requires recalibration each time it is worn. By contrast, our BP estimation model calculates continuous BP values without the need for peak and trough values of the measured signal, which eliminates the influence of different backpressure levels on BP estimation accuracy (**Extended Fig. 8 a-d**).

From the point of interfacial stability and wearability, the pressure sensor in the reference was embedded into a wristwatch, and the sensor needs to be well contacted with the skin upon the radial artery to facilitate satisfactory signal quality. However, the strap needs to be tied very tightly to ensure a good contact between the pressure sensor

and the skin upon the radial artery as the artery is located at the corner of the wrist, which will greatly reduce the weldability of the system. In contrast, the whole system in our work was encapsulated into a silicone wristband, providing a soft and user-friendly interface. Moreover, the active backpressure adaption module can precisely provide a powerful backpressure to the pressure sensors, maximizing its wearability while improving signal quality.

Modifications: We added these two references in **Supplementary Table 1** to summarize their technical details. Moreover, we added a detailed discussion on wireless wearable continuous BP monitoring techniques with these two recent references included to compare them with our system.

In Line 23, supplementary, we modified the text as “**Supplementary Note 1: Comparison of BP monitoring technologies.**”

Conventionally, non-invasive blood pressure is measured with a manual cuff, in which a pressure gauge is equipped to measure the air pressure inside the cuff. By inflating the manual cuff to a pressure that is higher than BP, systolic BP and diastolic BP can be recorded with the assistance of the stethoscope. While continuous BP measurements typically associate with invasive methods. Therefore, it's extremely important to develop wearable devices for continuous BP monitoring. To date, there are mainly four categories of technologies that are developed for continuous BP monitoring (Supplementary Table 1, Supplementary Fig. 1): (1) Optical based technology, i.e., Photoplethysmography (PPG) measures the changes in reflected light caused by volumetric variation of blood circulation for the estimation of BP; (2) Acoustic based technology, i.e., Ultrasound wall-tracking derives the changes in artery diameter by analyzing the echo signals from anterior vessel wall and posterior vessel wall to calculate continuous BP; (3) Electrical based technology, i.e., Electrodes array measures the variation of bioimpedance generated by blood propagation to predict BP; (4) Pressure sensor based technology, i.e., High precise pressure sensor, as presented in this work, are adopted to detect dynamic pressure from artery generated by blood propagation for the estimation of BP. Among them, PPG as one of the most commonly used methods, however, suffers from insufficient penetration depth, and thus are commonly used to measure the blood flow change in peripheral blood vessels, such as those in fingertips or earlobe. In comparison, Acoustic base technology is considered as the promising candidate for continuous BP monitoring due to its high penetration depth and robust sensing capability in hemodynamic parameters. Recently, totally wearable ultrasound system has been developed for BP monitoring and tissue imaging¹. However, for continuous BP monitoring application, ultrasound wall tracking relies on the isolated DBP and SBP to calculate the vascular resistance and stiffness related coefficient $\beta^{2,3}$ to calibrate the device. Therefore, personalized calibration process is required for each user, in which well-trained professional must be involved, and thus increase the usage barrier. Besides, high power consumption of the electrical circuit in high frequency (over 2 MHz) signal sampling and transmission increases the system bulkiness due to the need of bulky battery for long lifetime. Electrical based technology

utilities the impedance changes during blood circulations to measure pulse wave, which needs only a set of electrodes array applied upon the arteries. Although ultrathin graphene tattoo electrodes have been developed for bioimpedance measurement², bulky interface between electrodes and circuit, and complicated modulation and demodulation process of the sensing signal block its application in wireless and continuous BP monitoring. In comparison, pressure sensor based technology shows advances in device stability, signal processing and transmission circuit. Real wearable pressure sensor based system has been developed^{3,4} for wireless, continuous BP measurement. However, there are two mainly two drawbacks in pressure sensor based systems. The first one is BP calibration, that is converting the measured continuous pulse wave into continuous BP. Recent works^{3,4} adopted a transfer coefficient to map the peak and valley value of the sensor output into SBP and DBP, respectively. However, the output voltage of the pressure sensor is highly related to individual mechanical properties, such as vascular modulus and skin modulus, and experimental settings, such as back pressure level provided by the holder, which means repeated calibration process are required for each user even two times measurement for the same user due to the contribution of back pressure variation on output voltage. The second one is the poor interfacial performance between the pressure sensor and human skin. The pressure sensor needs to be tightly mounted on the skin to effectively detect the deformation caused by blood propagation, which puts high requirement on sensor encapsulation and system integration because high integration level will reduce sensor deformation in response to arterial deformation, greatly reducing measured signal quality. Strategies have been proposed to address the poor interfacial performance issue and increase system integration level, e.g., utilizing a wrist band or watch strap to provide powerful support to the pressure sensor, which, however, will significantly reduce user comfort due to high pressure level applied on the wrist.”

Supplementary Table 1. Technical comparison of continuous BP monitoring.

Method	Wireless? Totally wearable?	Continuous?	Accuracy				Device dimension
			DBP (mmHg)		SBP (mmHg)		
			ME	SD	ME	SD	
PPG ¹³	Yes PPG sensor and hard sampling board in a watch shaped wrist.	Not	-0.07±4.47		0.00±3.61		Not reported.
PPG ¹⁰	No Commercial PPG sensor and ECG electrode required.	Yes Beat to beat. Static BP.	3.23±4.75		4.43±6.09		Not reported.
PPG ¹⁴	Yes	Yes 1000s long. Continuous. Static BP.	Not reported.		0.24	1.18	Device size 40 mm × 20 mm Board size Not reported.
PPG ¹⁵	Yes Neither the sensor	Not	2.12	0.26	2.94	0.72	Not reported.

	nor the electrical circuit are designed in wearable format.							
Piezoelectric ultrasound transducer ¹⁶	No Connecting to sampling equipment required.	Yes 30s long. Continuous. Static BP.			Not reported.			23 mm × 20 mm
Piezoelectric ultrasound transducer ¹	Yes Yes	Yes Static BP			0.17 ± 4.92			Device size not reported. 20.29 cm ³ battery for 12 h lifetime.
Resistive pressure sensor combined with ECG ¹²	Yes Wearable pulse sensor and ECG electrodes are worn separately.	Yes 4 hour long. 15s time window. Static BP.	0.24	5.19	0.07	9.66		ECG patch 70 mm × 20 mm Pulse sensor 40 mm × 17 mm
Piezoelectric pressure sensor ⁴	No Sensor in wearable format. Rigid sampling board. Sensor and electrical board were not integrated.	Yes Static BP.			Not reported.		Not reported.	Device size 3.5 mm in diameter. Board size not reported.
Piezoelectric pressure sensor ⁵	Yes Yes	Yes Static BP.	-0.89	6.19	-0.32	5.28		Device size not reported. Integrated in a wristwatch. Watch size not reported.
Capacitive pressure sensor ¹⁷	No Connecting to sampling equipment required.	Yes 10 min. long. Static BP	0.48	1.96	1.43	1.96		8 mm × 8 mm
Capacitive pressure sensor ¹⁸	No Fixing wristband and sampling equipment required.	Yes 9 min. long. 70 beats window. Static BP			-0.054 ± 2.09			15 mm × 2 mm
Commercial silver electrodes. Bio-impedance ¹¹	No Bulky wristband and sampling board required.	Yes 60 min. long. 5-10s time window. Dynamic BP	-1.3	6	3.7	8.5		Device size 64 mm × 46 mm Board size not reported.
Graphene tattoo ² Bio-impedance	No Extra connecting wires required from the graphene tattoo to the sampling board.	Yes 5+ hours long. 5-10s time window. Dynamic BP	0.2	4.5	0.2	5.8		200 nm in thickness. Extra sampling board required.
Piezoelectric pressure sensor ★	Yes Soft sensor together with flexible sampling board encapsulated in a thin wristband.	Yes All day. Beat by beat. Dynamic BP	0.11	3.68	-0.05	4.61		Planar size: 150 mm × 350 mm Thickness 4 mm

★ This work.

Dynamic BP and static BP are defined as whether BP fluctuations were created during the measurement.

Newly added references:

1. Lin, M. et al. A fully integrated wearable ultrasound system to monitor deep tissues in moving subjects. *Nat Biotechnol* 1–10 (2023) doi:10.1038/s41587-023-01800-0.
2. Palombo, C. & Kozakova, M. Arterial stiffness, atherosclerosis and cardiovascular risk: Pathophysiologic mechanisms and emerging clinical indications. *Vascular Pharmacology* 77, 1–7 (2016).
3. Wada, T., Fujishiro, K., Fukumoto, T., Yamazaki, S. & Wada, T. Relationship Between Ultrasound Assessment of Arterial Wall Properties and Blood Pressure. *Angiology* 48, 893–900 (1997)
5. Min, S. et al. Clinical Validation of a Wearable Piezoelectric Blood-Pressure Sensor for Continuous Health Monitoring. *Advanced Materials* n/a, 2301627.

Common 3: Serious issue is that the revised manuscript claimed that total of 87 volunteers for blood pressure measurement, adding them in Extended Data. However, the main MS described only the accuracies of blood pressures in the initial 17 participants of previous version.

Then why did the author add 87 volunteers?

Current manuscript may lead readers to misunderstand the measurement condition.

Our response: We thank the referee for these comments. We are sorry if we didn't clearly describe this part in the first round revision. First, the data of 87 volunteers was collected for responding your first round comment, **as your professional suggestion of “a minimum of 85 subjects using an auscultatory sphygmomanometer as a reference”**. To facilitate better understanding, we reorganized **Fig. 4** and presented the test results on 87 volunteers after the static and dynamic performance validation of the TSMS. While for the selected 17 volunteer, we aimed to preliminarily study if biological aging or body mass index (BMI) will contribute to hypertension. From the results in **Fig. 4h**, the BP shows a slightly increasing trend with increased age, which a borderline hypertension individual with BMI over 24 (**Fig. 4i**) was observed in the youth group, indicating the borderline hypertension might be associated with the high level of BMI. To clarify this issue, we put comments in Line 394 of manuscript.

Modifications: In Line 364, manuscript, we added the description of the performance validation of the TSMS on 87 volunteers and modified the text as “To validate the measurement robustness of the TSMS, dynamic evaluation process was conducted. The evaluation is associated with wearing TSMS on the right wrist to continuously measure BP, and at the meantime commercial equipment (BioPAC) equipped with CNAP monitor, which has been proved providing clinically acceptable accuracy⁶⁵, attached on the same volunteers to measure reference BP, as illustrated in **Fig. 4a** and **Supplementary Video 2. Fig. 4b** shows the comparison between BP patterns measured

by the TSMS and the BioPAC of a volunteer, in which up to 25 minutes of continuous BP data is measured by the TSMS and agrees very well with those measured by BioPAC, indicating the excellent performance of the TSMS. Besides, the statistical comparison between the TSMS and BioPAC also shows excellent agreement, especially in DBP, with the same mean value as well as a slightly smaller dynamic range in 25 minutes (**Fig. 4c, Supplementary Fig. 20**). To further study the capability of our TSMS in dynamic monitoring of BP, hand grip and cold pressor process (HGCP) cycles, which has been proved an efficient way to increase BP due to the triggered sympathetic response^{8,66}, was adopted to realize a large dynamic range of BP (**Fig. 4d**). During HGCP maneuvers, the volunteer exercised with the hand grip for 2 minutes to slowly elevate his BP gradually before immersing his hand into a bucket of ice-cold water (4 °C) for 1 minute to further raise BP. Then, a resting period for 5 minutes allows BP to drop to a low level. Bicycle HGCP maneuvers were conducted to raise a maximum SBP and DBP of 163 mmHg and 116 mmHg, respectively (**Fig. 4e, gray dotted line**). The continuous BP patterns measured by the TSMS show satisfying agreement, with the predicted maximum SBP and DBP at 158 mmHg and 109 mmHg, respectively. **Fig. 4f** shows the statistical comparison between the measured BP and reference BP, which further proves the excellent agreement between the measured BP and reference BP, with negligible error in mean value of SBP and DBP although a slightly smaller dynamic distribution (**Supplementary Fig. 21**). To validate the measure accuracy of the TSMS system, a total number of 87 volunteers were selected to continuously monitor their BP for 2 minutes with BP measured by commercial noninvasive arterial blood pressure monitor (CNAP) and commercial sphygmomanometer (Omron HEM-7156T) as references. **Fig. 4g** shows the statistic error distribution of the 87 volunteers with CNAP as a reference, where most of error bars are less than 10 mmHg regardless of SBP or DBP. Moreover, quantitative statistics on SBP and DBP show that excellent measurement accuracy was achieved with the TSMS, with over 70% for error < 5 mmHg and over 98% for error < 15 mmHg (**Supplementary Fig. 22**).

Hypertension is a highly prevalent condition with numerous health risks that could be caused by various factors. Among those pathogenic factors, biological aging and body mass index (BMI) have been reported showing positive association with BP^{67,68}. To study how the age and BMI affect BP and further validate the TSMS as a reliable continuous BP monitoring system, we select 17 volunteers, with comprehensive age and BMI, to perform a pilot study to monitor their daily BP with TSMS and take BP measured with a commercial sphygmomanometer (Omron HEM-7156T), the gold standard for noninvasive BP measurement, as a reference (**Supplementary Fig. 23**). The volunteers were grouped into 4 teams based on their age and 3 teams based on their body mass index (BMI), respectively. The BP in four age groups, teenager (<20), youth (20-35), middle-aged (35-50) and elderly (>50), is presented in **Fig. 4h** and **Supplementary Fig. 24**, where a slightly increasing trend can be found in BP with the increased age regardless of gender. While there is a borderline hypertension individual in the age group of youth, which indicates the multiple pathogenic factors of hypertension. Besides, a hypertension individual with SBP over 150 mmHg and DBP over 100 mmHg was also recorded. To further study the correlation between BP and

BMI and figure out the possible causative factors of the two hypertension individuals, the volunteers were grouped into slim ($BMI < 20$), normal ($20 < BMI < 24$) and overweight ($BMI > 24$), based on their BMIs. **Fig. 4i** presents the statistical plots of the BP of the 17 volunteers, where the borderline hypertension individual locates in the overweight group, indicating the hypertension could be associated with high levels of BMI. Nevertheless, the high BP of the borderline hypertension individual could be caused by other factors since the rest of volunteers in the overweight group show normal BP. Besides, there is no obvious correlation between BP and BMI observed due to the limited sample size.”

Figure 4. Performance evaluation of the wireless wristband for continuous BP monitoring. (a). Schematic illustration of continuous BP monitoring in an office scene with the wireless wristband. (b). Performance validation of predicted SBP and DBP in comparison with commercial continuous BP monitoring, Bio-PAC system, for 25 minutes continuous measurement. (c). Statistical plots present the comparison between our wireless wristband and Bio-PAC system in SBP and DBP. (d). Schematic illustration of HGCP process for producing a wide range of dynamic BP. Volunteer's hand is immersed in cold water (4°C) for 2 minutes, followed by hand grip exercise for 2 minutes to increase BP. (e). Predicted SBP and DBP in comparison with Bio-PAC during three HGCP cycles. (f). Statistical plots present the comparison between our wireless wristband and Bio-PAC system in SBP and DBP during HGCP cycles. (g). Violin plots representing the BP measurement accuracy of the TSMS compared with commercial CNAP in a total number of 87 volunteers. (h) and (i). Statistical comparison of BP measured by the wireless wristband and measured by commercial cuff-based monitor in a total number of 17 volunteers, divided into 4 age groups (h) and 3 BMI groups (i).

Supplementary Fig. 22. Quantitative statistics presenting the measurement accuracy of the TSMS on a total number of 87 volunteers. (a) and (b). Measurement accuracy comparison in DBP (a) and SBP (b) with commercial sphygmomanometer as a reference. Measurement accuracy statistics on systolic BP (c) and diastolic BP (d) representing the measurement accuracy in different BP categories under the standard of America Hypertension Association (AHA) with continuous BP measured by CNAP as reference.

Response to comments of Referee #3

Comments from Referee #3:

As a system the reported work is very impressive. Most of previous review comments were very well responded. One more comment, authors may conduct a thorough survey on wearable blood pressure monitoring techniques including wearable ultrasound for blood pressure monitoring, etc.

Our response: We thank the referee for the valuable comments. We conducted a thorough review on wearable BP monitoring techniques and summarized the technical comparison in **Supplementary Table 1**. Besides, we added a detailed discussion on current techniques for continuous BP monitoring in **Supplementary Note 1**.

Modifications: In Line 23, supplementary, we added a detailed discussion on wearable BP monitoring techniques and modified the text as “**Supplementary Note 1: Comparison of BP monitoring technologies**”.

Conventionally, non-invasive blood pressure is measured with a manual cuff, in which a pressure gauge is equipped to measure the air pressure inside the cuff. By inflating the manual cuff to a pressure that is higher than BP, systolic BP and diastolic BP can be recorded with the assistance of the stethoscope. While continuous BP measurements typically associate with invasive methods. Therefore, it's extremely important to develop wearable devices for continuous BP monitoring. To date, there are mainly four categories of technologies that are developed for continuous BP monitoring (Supplementary Table 1, Supplementary Fig. 1): (1) Optical based technology, i.e., Photoplethysmography (PPG) measures the changes in reflected light caused by volumetric variation of blood circulation for the estimation of BP; (2) Acoustic based technology, i.e., Ultrasound wall-tracking derives the changes in artery diameter by analyzing the echo signals from anterior vessel wall and posterior vessel wall to calculate continuous BP; (3) Electrical based technology, i.e., Electrodes array measures the variation of bioimpedance generated by blood propagation to predict BP; (4) Pressure sensor based technology, i.e., High precise pressure sensor, as presented in this work, are adopted to detect dynamic pressure from artery generated by blood propagation for the estimation of BP. Among them, PPG as one of the most commonly used methods, however, suffers from insufficient penetration depth, and thus are commonly used to measure the blood flow change in peripheral blood vessels, such as those in fingertips or earlobe. In comparison, Acoustic base technology is considered as the promising candidate for continuous BP monitoring due to its high penetration depth and robust sensing capability in hemodynamic parameters. Recently, totally wearable ultrasound system has been developed for BP monitoring and tissue imaging¹. However, for continuous BP monitoring application, ultrasound wall tracking relies on the isolated DBP and SBP to calculate the vascular resistance and stiffness related

coefficient β to calibrate the device. Therefore, personalized calibration process is required for each user, in which well trained professional must be involved, and thus increase the usage barrier. Besides, high power consumption of the electrical circuit in high frequency (over 2 MHz) signal sampling and transmission increases the system bulkiness due to the need of bulky battery for long lifetime. Electrical based technology utilizes the impedance changes during blood circulations to measure pulse wave, which needs only a set of electrodes array applied upon the arteries. Although ultrathin graphene tattoo electrodes have been developed for bioimpedance measurement², bulky interface between electrodes and circuit, and complicated modulation and demodulation process of the sensing signal block its application in wireless and continuous BP monitoring. In comparison, pressure sensor based technology shows advances in device stability, signal processing and transmission circuit. Real wearable pressure sensor based system has been developed^{3,4} for wireless, continuous BP measurement. However, there are two mainly two drawbacks in pressure sensor based systems. The first one is BP calibration, that is converting the measured continuous pulse wave into continuous BP. Recent works^{3,4} adopted a transfer coefficient to map the peak and valley value of the sensor output into SBP and DBP, respectively. However, the output voltage of the pressure sensor is highly related to individual mechanical properties, such as vascular modulus and skin modulus, and experimental settings, such as back pressure level provided by the holder, which means repeated calibration process are required for each user even two times measurement for the same user due to the contribution of back pressure variation on output voltage. The second one is the poor interfacial performance between the pressure sensor and human skin. The pressure sensor needs to be tightly mounted on the skin to effectively detect the deformation caused by blood propagation, which puts high requirement on sensor encapsulation and system integration because high integration level will reduce sensor deformation in response to arterial deformation, greatly reducing measured signal quality. Strategies have been proposed to address the poor interfacial performance issue and increase system integration level, e.g., utilizing a wrist band or watch strap to provide powerful support to the pressure sensor, which, however, will significantly reduce user comfort due to high pressure level applied on the wrist.”

Supplementary Table 1. Technical comparison of continuous BP monitoring.

Method	Wireless? Totally wearable?	Continuous?	Accuracy				Device dimension
			DBP (mmHg)		SBP (mmHg)		
			ME	SD	ME	SD	
PPG ¹³	Yes PPG sensor and hard sampling board in a watch shaped wrist.	Not	-0.07±4.47		0.00±3.61		Not reported.
PPG ¹⁰	No Commercial PPG sensor and ECG electrode required.	Yes Beat to beat. Static BP.	3.23±4.75		4.43±6.09		Not reported.
PPG ¹⁴	Yes	Yes 1000s long. Continuous.	Not reported.		0.24	1.18	Device size 40 mm ×

		Static BP.						20 mm Board size Not reported.
PPG ¹⁵	Yes Neither the sensor nor the electrical circuit are designed in wearable format.	Not	2.12	0.26	2.94	0.72		Not reported.
Piezoelectric ultrasound transducer ¹⁶	No Connecting to sampling equipment required.	Yes 30s long. Continuous. Static BP.			Not reported.			23 mm × 20 mm
Piezoelectric ultrasound transducer ¹	Yes Yes	Yes Static BP			0.17 ± 4.92			Device size not reported. 20.29 cm ³ battery for 12 h lifetime.
Resistive pressure sensor combined with ECG ¹²	Yes Wearable pulse sensor and ECG electrodes are worn separately.	Yes 4 hour long. 15s time window. Static BP.	0.24	5.19	0.07	9.66		ECG patch 70 mm × 20 mm Pulse sensor 40 mm × 17 mm
Piezoelectric pressure sensor ⁴	No Sensor in wearable format. Rigid sampling board. Sensor and electrical board were not integrated.	Yes Static BP.		Not reported.		Not reported.		Device size 3.5 mm in diameter. Board size not reported.
Piezoelectric pressure sensor ³	Yes Yes	Yes Static BP.	-0.89	6.19	-0.32	5.28		Device size not reported. Integrated in a wristwatch. Watch size not reported.
Capacitive pressure sensor ¹⁷	No Connecting to sampling equipment required.	Yes 10 min. long. Static BP	0.48	1.96	1.43	1.96		8 mm × 8 mm
Capacitive pressure sensor ¹⁸	No Fixing wristband and sampling equipment required.	Yes 9 min. long. 70 beats window. Static BP			-0.054 ± 2.09			15 mm × 2 mm
Commercial silver electrodes. Bio-impedance ¹¹	No Bulky wristband and sampling board required.	Yes 60 min. long. 5-10s time window. Dynamic BP	-1.3	6	3.7	8.5		Device size 64 mm × 46 mm Board size not reported.
Graphene tattoo ² Bio-impedance	No Extra connecting wires required from the graphene tattoo to the sampling board.	Yes 5+ hours long. 5-10s time window. Dynamic BP	0.2	4.5	0.2	5.8		200 nm in thickness. Extra sampling board required.
Piezoelectric	Yes Soft sensor	Yes All day.	0.11	3.68	-0.05	4.61		Planar size: 150 mm

pressure sensor ★	together with flexible sampling board encapsulated in a thin wristband.	Beat by beat. Dynamic BP	× 350 mm Thickness 4 mm
--	-----------------------------	-------------------------------

★ This work.

Dynamic BP and static BP are defined as whether BP fluctuations were created during the measurement.

Newly added references:

1. Lin, M. et al. A fully integrated wearable ultrasound system to monitor deep tissues in moving subjects. *Nat Biotechnol* 1–10 (2023) doi:10.1038/s41587-023-01800-0.
5. Min, S. et al. Clinical Validation of a Wearable Piezoelectric Blood-Pressure Sensor for Continuous Health Monitoring. *Advanced Materials* n/a, 2301627.

REVIEWERS' COMMENTS

Reviewer #2 (Remarks to the Author):

The authors have addressed all questions and it can be accepted at its current form.

Reviewer #3 (Remarks to the Author):

Previous review comment was well responded. Thanks!